

# Simulating the impact of an AMOC weakening on the Antarctic Ice Sheet using a coupled climate and ice-sheet model

Anna Höse[1,2,3], Moritz Kreuzer[1,3], Willem Huiskamp[1], Stefan Petri[1], and Georg Feulner[1,3]

[1]Potsdam-Institute for Climate Impact Research (PIK), Member of the Leibniz Association, Potsdam, Germany
[2]Alfred-Wegener-Institut, Helmholtz-Zentrum für Polar- und Meeresforschung, Potsdam, Germany
[3]Institute of Physics and Astronomy, University of Potsdam, Potsdam, Germany

**Correspondence:** Anna Höse (anna.hoese@awi.de)

**Abstract.** Climate model studies show that a shutdown of the Atlantic Meridional Overturning Circulation (AMOC) reduces northward heat transport into the North Atlantic, which causes an accumulation of heat in the sub-tropical Southern Ocean. The Antarctic Ice Sheet meanwhile has been shown to be particularly susceptible to temperature changes in ocean water flowing into the cavities of its grounded ice shelves. How AMOC-induced modulation of inter-hemispheric heat transport could influence the present-day state of the Antarctic Ice Sheet via a southward propagation of warm anomalies is little studied. As both, the AMOC as well as the West Antarctic Ice Sheet, are classified as climate tipping points, which can trigger irreversible changes in the Earth System, it is highly relevant how both systems interact with each other.

In this study, we simulate for the first time a shutdown of the AMOC in a global climate model interactively coupled to an ice-sheet model for Antarctica. In line with previous studies, an AMOC shutdown causes increased sea-surface temperatures in the Southern Hemisphere along with a small shift in the mid-latitude westerlies. However, Southern Ocean subsurface temperatures, which drive basal melt in Antarctica, do not change in most regions along the Antarctic margin for the first eight centuries post AMOC shutdown. Therefore, we do not find a change in the total Antarctic Ice volume in this time span. At later times, this is followed by a shift towards stronger Ross Sea convection, causing negative subsurface temperature anomalies of $-1.4\,°C$ on average. This cooling decreases basal melt in Antarctica, however increased calving balances the ice mass change. Even though our coupling approach strongly simplifies eddy mass and heat fluxes in the Southern Ocean, and does not resolve flows within ice-shelf cavities, our approach is an important first step to systematically investigate Earth-system stability in coupled climate–ice-sheet models.

## 1 Introduction

The Atlantic Meridional Overturning Circulation (AMOC) is considered a tipping element of the Earth system, which can undergo significant and often irreversible changes once a critical threshold is crossed (Manabe and Ronald, 1988; Rahmstorf et al., 2005; McKay et al., 2022). AMOC strength could substantially reduce or collapse under changing climate conditions such as increased global mean surface temperature (GMST) or enhanced North Atlantic freshwater input due to increased precipitation or runoff/ ice-sheet discharge. As part of the global thermohaline ocean circulation, the AMOC plays a key role in transporting heat from the Southern to the Northern Hemisphere (Rahmstorf, 2002; Kuhlbrodt et al., 2007; Feulner et al.,



2013) and is thereby crucial for global climate regimes (Vellinga and Wood, 2002; Knutti et al., 2004; Stouffer et al., 2006; Timmermann et al., 2007; Jackson and Wood, 2018) as well as present-day climate conditions in Europe (Jackson et al., 2015; Meccia et al., 2023). Existing observational data of the AMOC strength since 2004 show a downward trend (Smeed et al., 2018; Mishonov et al., 2024), and studies based on other indicators such as sea-surface temperature or climate proxy records (Dima and Lohmann, 2010; Caesar et al., 2021) are able to reproduce this trend. Suggested drivers of decreasing AMOC

strength are increasing freshwater input in the North Atlantic due to more precipitation, sea-ice loss, as well as melt water from the Greenland ice sheet (Caesar et al., 2018). Additionally, climate projections using different state-of-the-art climate models confirm a continuing slowdown of the circulation at least until 2100 (IPCC AR6 WG1 Ch. 9.2.3.1, Fox-Kemper et al., 2021; Golledge et al., 2019; Gierz et al., 2015), which happens partly due to GMST increase, partly due to more freshwater input in higher latitudes (Pontes and Menviel, 2024). Different studies warn that there is a severe risk of AMOC tipping in the

next century (Ditlevsen and Ditlevsen, 2023; van Westen et al., 2024) and that models tend to overestimate AMOC stability (Hofmann and Rahmstorf, 2009).

Past climate reconstructions based on ice and marine core data suggest that the AMOC has changed its state several times in Earth's history (Barbante et al., 2006; Du et al., 2020; Henry et al., 2016). One prominent example are Dansgaard-Oeschger (DO) events in the Quaternary, i.e. oscillations between stadial and interstadial conditions in Greenland, as determined from

temperature reconstructions based on ice core or sediment proxy data (Barbante et al., 2006; Voelker, 2002; Stocker and Johnsen, 2003; Landais et al., 2015). Varying AMOC strength (McManus et al., 2004), which changes the heat transport to Greenland, is the most consistent explanation for these climate shifts, as it can explain similar oscillations in ice-core proxies from Antarctica (Clark et al., 2002). The theory to explain the link between oscillations found in proxy data of both hemispheres is the antiphased bipolar seesaw (Broecker, 1998; Stocker and Johnsen, 2003). A weak AMOC reduces heat

export to the Northern Hemisphere (NH), which therefore accumulates in the South Atlantic, propagates to the Southern Ocean (SO) and impacts the surface temperature and, consequently, stratification there. Pedro et al. (2018) suggested, that the shifted timing of oscillations in temperature reconstructions from Antarctica compared to those extracted from proxies from Greenland (Barbante et al., 2006) results due to the inertia of heat accumulation in the southern Atlantic, especially at the surface. Additionally the weakening of the AMOC return flow might trigger open-ocean convection in the SO (Willeit et al.,

2025) which warms the atmosphere efficiently by deep ocean heat release (Pedro et al., 2016). Even though under present-day climate conditions open-ocean convection is hardly observed, a study by Skinner et al. (2020) proposes that open ocean convection was amplifying rapid temperature changes around 40k yrs before present as found in proxy records.

Climate models of different complexity support the seesaw theory by simulating an AMOC shutdown in hosing experiments which prescribe an artificial freshwater flux to the North Atlantic. This freshwater represents e.g. high meltwater input from

Greenland or increased precipitation rates and results in a reduction or collapse of the AMOC as it suppresses convection by increasing the water stratification. Using this experiment setup, several studies (e.g. Stouffer et al., 2006; Jackson and Wood, 2018; Diamond et al., 2025) have investigated the consequences of a collapsed AMOC. They consistently show a compensation of oceanic heat transport by a southward shift of the atmospheric Hadley circulation and the intertropical convergence zone (ITCZ) (Cheng et al., 2007; Kageyama et al., 2013; Knutti et al., 2004; Orihuela-Pinto et al., 2022; Tierney et al., 2008). As



a response to the shifted ITCZ, the strength of the mid-latitude westerlies above the SO increases (Lee et al., 2011). Pedro et al. (2018) show, under 19 ka BP climate conditions, that after an AMOC collapse most heat accumulates in the interior ocean north of the Antarctic Circumpolar Current (ACC), increasing the heat reservoir in the South Atlantic as proposed by Stocker and Johnsen (2003). Additionally, they find an increase in heat content of the Indian and Pacific ocean. The zonally closed ACC isolates the SO and only allows for slow southward oceanic heat propagation, as no Kelvin or Rossby wave

propagation is possible, only eddy flux processes (Pedro et al., 2018). However, the consequences of an AMOC collapse south of the ACC and especially on the Antarctic Ice Sheet (AIS) are comparatively less researched than Northern Hemisphere impacts. A recent study by Berdahl et al. (2024) first addressed drivers for changes in the temperature profile of the SO during periods of strongly reduced AMOC strength. According to that study, even though there are increasing *surface* temperatures, *subsurface* temperatures at the depths of the Antarctic ice shelf cavities are decreasing. These might be crucial for interactions

with the AIS, although feedbacks between SO and the AIS are not investigated in their study, because their model lacks – as most climate models do – an interactive AIS component. Whereas atmospheric signals after an AMOC collapse propagate in less than a century, it remains unclear if changes in SO deep water formation due to the bipolar seesaw effect add additional thermohaline anomalies in regions south of the ACC.

Model studies are crucial to understand the interactions between the atmosphere, ocean, sea-ice, and ice sheet in Antarctica,

as available datasets to investigate large-scale processes and feedbacks in the SO are limited due to its remote location and/or short time series. Satellite data from 1992 to 2020 show increasing rates of ice mass loss of the Antarctic Ice Sheet, summing up to a global sea level contribution of $7.4 \pm 1.5 \, \mathrm{mm}$ (Otosaka et al., 2023). Enhanced basal melt rates in Antarctica particularly threaten the stability of the West Antarctic Ice Sheet (WAIS), whose tipping point is estimated to be reached at an oceanic warming level of 1.5 °C compared to preindustrial times (Garbe et al., 2020). AIS projections forced with data of CMIP5

models (Seroussi et al., 2020) expect accelerating ice loss in Antarctica in the next 80 years and a study by Naughten et al. (2023) conclude that increasing melt rates of the WAIS until at least 2100 are unavoidable. How this melt water will feedback to ocean behaviour is still uncertain, nevertheless studies have shown that the interactions between these climate components has implications for the overall climate system (Bronselaer et al., 2018; Golledge et al., 2019).

As it is likely that sub-systems of the climate system as the AMOC and the WAIS are strongly interrelated, a review by Wun-

derling et al. (2023) summarized the current knowledge about how tipping points can stabilize or destabilize each other. Their study shows that researchers often assume a positive feedback of an AMOC collapse to the WAIS due to higher sea-surface temperatures around the AIS. This assumption is also used to set up a statistical model that assesses the probabilities for tipping point cascades (Wunderling et al., 2021). Due to the current declining trend of the AMOC strength and melting ice sheets on both hemispheres, it is crucial to understand interactions between the different Earth system components. Nevertheless, the

impact of an AMOC collapse on the Antarctic Ice Sheet has never, to the best of our knowledge, been investigated by a physical coupled model, that is able to capture the rather slow response times of ice sheets on centennial to millennial time-scales. Therefore, we set up a coupled climate–ice sheet model simulation to investigate the impact of an AMOC collapse on the AIS in a freshwater hosing experiment. This approach, that couples a climate model with an Antarctic Ice Sheet model via the ice-ocean interface, is used to address the question: How do melt water fluxes from the AIS change during an AMOC shutdown on





centennial to millennial timescales? Furthermore we investigate whether heat accumulation in the Southern Hemisphere (SH) increases AIS mass loss via an increase in basal melt. To understand changes in the ice-sheet volume, we analyse hydrographic changes in the SO and identify main drivers of Antarctic mass balance changes.

## 2  Methods

### 2.1  Model description

We employ a modified version of the CM2Mc Earth System Model (ESM) of Galbraith et al. (2011) coupled to the Parallel Ice Sheet Model (PISM) v1.0 (Bueler and Brown, 2009; Winkelmann et al., 2011; Garbe et al., 2020) via the offline coupling framework described in Kreuzer et al. (2021). CM2Mc consists of the atmosphere model AM2.1, the land model LandLAD, the Modular Ocean Model version 5 (MOM5), and the dynamical Sea-Ice Simulator (SIS). They are coupled by the Flexible Modeling System (FMS) (Delworth et al., 2006). The atmosphere grid has a latitudinal resolution of $3°$ and a longitudinal resolution of $3.75°$, with 24 vertical levels. MOM5 utilises an Arakawa B-grid in a tri-polar configuration (Galbraith et al., 2011). Its lateral grid resolution is nominally $3°$, varying latitudinally to a minimum of $0.6°$ at the equator to better resolve equatorial dynamics. The vertical grid has 28 layers implemented in the rescaled pressure ($p^*$) coordinate. Layer thickness varies between $10\,\mathrm{dbar}$ at the surface and $506\,\mathrm{dbar}$ in the deep ocean (Griffies, 2014).

PISM is used to simulate the AIS on a cartesian grid with a horizontal resolution of $16\,\mathrm{km} \times 16\,\mathrm{km}$. In the vertical, the grid
spacing ranges from $20\,\mathrm{m}$ at the ice base to $100\,\mathrm{m}$ for the thickest ice domes. Ice velocities are calculated by a superposition of the shallow-ice approximation and shallow-shelf approximation of the Stokes flow. We use an adopted version of PISM v1.0, which includes a precipitation scaling as introduced by Garbe et al. (2020). The scaling increases precipitation with decreasing ice elevation in order to account for the moisture holding capacity of air dependent on its temperature. The surface mass balance (surface melt and runoff) is computed using a positive degree-day (PDD) scheme. Melt coefficients are set to $3\,\mathrm{mm}$ per
PDD for snow and $9\,\mathrm{mm}$ per PDD for ice. The Glen-Paterson-Budd-Lliboutry-Duval flow law describes the ice rheology in the thermomechanically coupled model with a freely evolving three-dimensional enthalpy field (Aschwanden et al., 2012). Basal shear stress is parameterized dependent on the basal velocity and the yield stress that results from the Mohr-Coulomb criterion (Garbe et al., 2020; Cuffey and Paterson, 2010). Calving is implemented by the eigencalving approach by Levermann et al. (2012). Additionally, a subgrid scheme that captures calving fronts for different shelf geometries is used (Albrecht et al., 2011)
and a minimum thickness criterion of $50\,\mathrm{m}$ at the calving front is applied. The adaptive time-stepping scheme (Bueler et al., 2007) reduces computational costs by choosing the maximum possible time step based on the internal dynamic state of the system. The source code of PISM is identical to the one used in Garbe et al. (2020), except for one bug fix on the approximation of the driving stress at floating ice margins that was committed later in PISM version 2.0.

Coupling is done offline using the framework of Kreuzer et al. (2021). It exchanges mass and energy fluxes between ocean
and ice sheet through the Potsdam Ice-shelf Cavity model (PICO) (Reese et al., 2018), which is implemented as a sub-module in PISM. PICO calculates sub-shelf melt rates by parametrising the vertical overturning circulation inside ice-shelf cavities. For that purpose it uses a box model based on Olbers and Hellmer (2010), but extended to two horizontal dimensions, that





divides the ocean ice-shelf boundary into 19 Antarctic basins (see Fig. 2 in Reese et al., 2018). In each basin, melting and freezing below the ice shelves is calculated based on ocean temperature and salinity at the depth of the continental shelf.

Figure 3 in Kreuzer et al. (2021) visualises the conceptual idea of the offline variable exchange between MOM5 and PISM. MOM5 and PISM are run sequentially for a fixed coupling time step of 10 years. After each step, the coupling framework processes the model outputs to make it compatible between the different model grids and then restarts the models. The oceanic fluxes to PISM are based on regridded temperature and salinity fields at the AIS margin of MOM5. From the 3D temperature and salinity fields, values at the depth corresponding to the mean continental shelf topography are extracted and horizontally

averaged, resulting in a scalar value for each basin (Kreuzer et al., 2021). As MOM5 shows warm biases around the Antarctic continent, anomalies relative to the last 500 years of the climate spinup (see section 2.2) are calculated and applied to the forcing data of PISM standalone runs (the basin mean values of these data are included in Fig. 2 in Reese et al., 2018). PISM provides basal mass, calving, and surface mass fluxes aggregated per basin as well as the enthalpy required to melt the ice. These are regridded and inserted into the ocean as freshwater and enthalpy fluxes. In contrast to Kreuzer et al. (2021), basal

mass fluxes are inserted not at the surface but at the calving front ice-draft depth (which is determined as the mean depth at the outermost PICO box) as this represents the vertical insertion of ice-shelf cavity meltwater more realistically.

A further minor modification to the coupling framework by Kreuzer et al. (2021) is the removal of river runoff fluxes in the Antarctic domain as precipitation runoff into the Southern Ocean is represented by the PISM ice-sheet instance instead. As PISM is not coupled to the atmosphere component of CM2Mc, we do not strictly conserve water there. However, discrepancies

are negligible, as we put artificial freshwater to the system in our experiments (i.e. hosing). We force the surface of PISM with climatological means of surface air temperature and precipitation which are described in the next section.

Most climate models use prescribed fluxes for Antarctic freshwater discharge, whereas our approach is able to capture changes in the dynamics of ice sheet and shelves in Antarctica and their interaction with the ocean. As a result, the different components of discharge from the Antarctic Ice Sheet into the surrounding ocean (surface runoff, basal melt, and calving)

evolve dynamically in correspondence to the applied ocean-to-ice forcing.

## 2.2 Experimental design and methods

The atmospheric boundary conditions for PISM are based on a multiple regression analysis of ERA-Interim data (Dee et al., 2011) resulting in a parameterisation of mean annual and mean summer surface air temperature as a function of latitude and surface elevation, using an atmospheric lapse rate of $\Gamma = -8.2\,°\mathrm{C}\,\mathrm{km}^{-1}$ (Albrecht et al., 2020). The mean precipitation field is

calculated as the average between 1986 and 2005 from the output of the Regional Atmospheric Climate MOdel (RACMOv2.3) published by van Wessem et al. (2018). For the PISM spinup, ocean forcing is provided by observational temperature and salinity data at the sea floor on the continental shelf of Antarctica averaged over the time period from 1975 to 2012 (Schmidtko et al., 2014). Anomalies in the coupled framework are applied to these oceanic fields.

CM2Mc runs with pre-industrial atmospheric conditions as well as land cover of the year 1860. For the radiative forcing

this implies a solar irradiance of $1364.67\,\mathrm{W}\,\mathrm{m}^{-2}$ and greenhouse gas concentrations as given by Delworth et al. (2006). Further conditions provided by monthly mean climatologies derived from reanalysis are described by Galbraith et al. (2011).



The parameter set in the example configuration CM2M_coarse_BLING as distributed with the MOM5 code does not entirely reproduce the results shown by Galbraith et al. (2011). Therefore, we modified several parameters (see Appendix B) to represent the pre-industrial climate state. Additionally, the CM2Mc time steps are reduced from $3\,\mathrm{h}$ to $1.5\,\mathrm{h}$ for the ocean and from $1.5\,\mathrm{h}$

to $0.75\,\mathrm{h}$ hours for the atmosphere.

The coupled system was spun-up in three stages. The first makes use of the ocean and sea-ice components of CM2Mc in standalone mode with prescribed atmospheric boundary conditions. It allows for an equilibration of the ocean with respect to a changed freshwater input of the Antarctic Ice Sheet. In this, the default river runoff values at the southernmost cells of the ocean were replaced by static freshwater fluxes calculated as the 1000-year mean of a PISM standalone spinup. This setup was run for

8000 years to avoid abrupt changes when coupling it to the interactive PISM. In the second spinup stage, CM2Mc was initialized using ocean and sea-ice restarts from stage 1 with the same PISM forcing and integrated for further 1000 years, now including a dynamic atmospheric component. Finally, once stage 2 was complete, PISM was coupled interactively and integrated for 500 years. All following experiments are integrated in this coupled configuration extended from the state of this last spinup. Throughout our analysis, we define the AMOC as the maximum zonally integrated meridional stream function strength in

the Atlantic between $20°\,\mathrm{N}$ and $60°\,\mathrm{N}$ below $500\,\mathrm{m}$ and the AABW circulation cell strength is defined as absolute value of the minimum global meridional overturning circulation (GMOC) below $2000\,\mathrm{m}$. The GMOC is defined as the aggregation of AMOC and the Indo-Pacific Meridional Overturning Circulation (PMOC) strength (maximum between $20°\,\mathrm{N}$ and $90°\,\mathrm{N}$ below $500\,\mathrm{m}$ in the Pacific basin). The stage 3 climate state has a maximum AMOC strength of $\approx 21\,\mathrm{Sv}$ (1 Sverdrup= $1\,\mathrm{Sv} = 10^6\,\mathrm{m}^2$) and a GMOC strength of $\approx 23\,\mathrm{Sv}$. The mean temperature and salinity values that are used as input for PISM oscillate around

$-0.95 \pm 0.3\,°\mathrm{C}$ and $34.575 \pm 0.025\,\mathrm{g\,kg^{-1}}$ during the stage 3 spinup.

The setup of the North Atlantic freshwater hosing experiment which we apply to the model follows the protocols of the North Atlantic Hosing Model Intercomparison Project (NAHos-MIP) of Jackson et al. (2023). We use a uniform distribution of freshwater with hosing flux of $0.3\,\mathrm{Sv}$ that is applied to the North Atlantic and Arctic Oceans by distributing it in the regions above $50°\,\mathrm{N}$ in the Atlantic and above the Bering Strait in the Pacific (see Fig. 1a in Jackson et al., 2023). We performed

both uniform hosing experiments using 0.1 (not shown here) and 0.3 Sv hosing strength, but show only results of the 0.3 Sv uniform hosing experiment here, as the weaker forcing does not lead to a full AMOC collapse in our model and therefore is not suitable to investigate our research question. The freshwater is added to the original river runoff values of the model and no mass restoring is applied. To prevent a rapid rebound of AMOC strength, we maintain the hosing for the entire duration of the experiment. A control experiment is run in parallel with the hosing run. Both runs are integrated for 1500 years. These two runs,

hereafter called HOSING and CONTROL, and their difference are the focus of this study. All comparisons are presented as the difference between the 100-year means of HOSING and CONTROL to avoid the evaluation of short term climate variability.





# 3 Results

## 3.1 Global response and average AIS changes

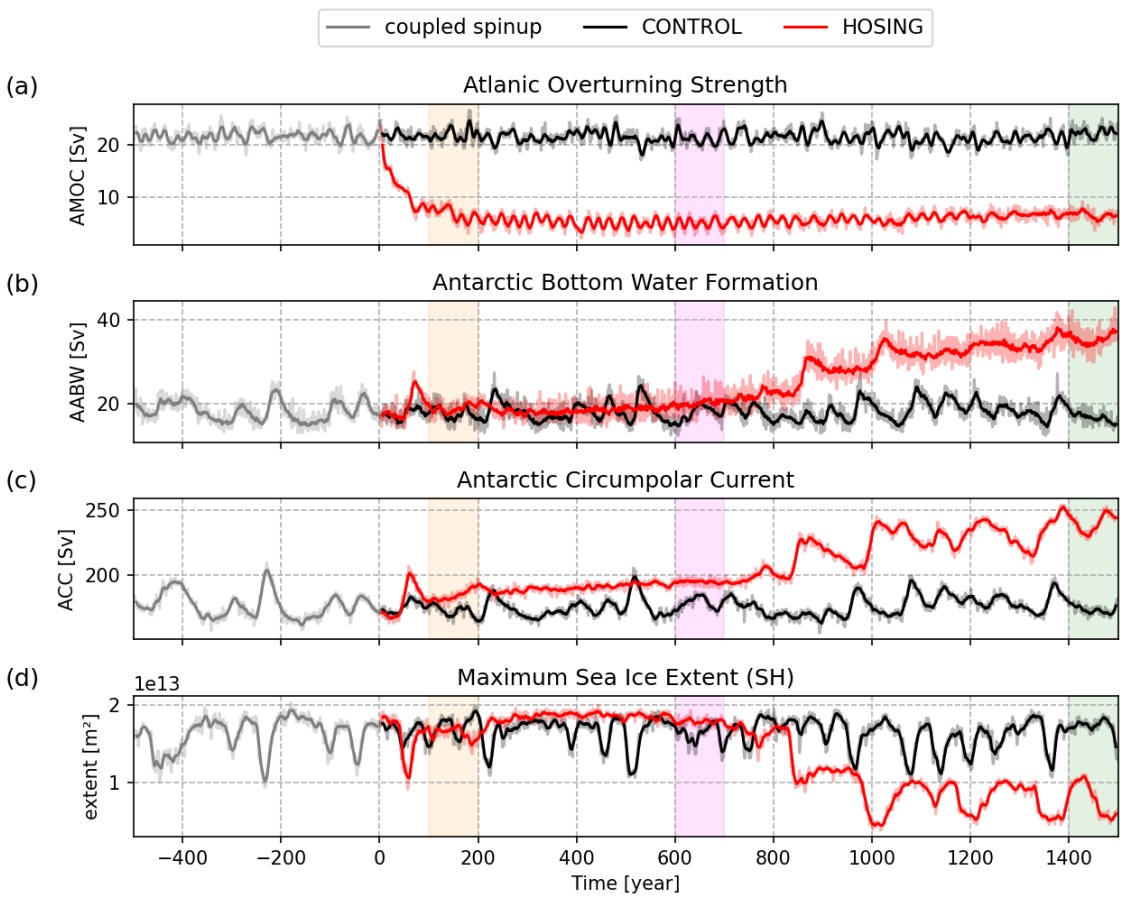

**Figure 1.** Time series of ocean diagnostics: (a) AMOC strength in Sv, (b) AABW formation in Sv, (c) ACC strength in Sv and (d) maximum sea-ice extent in the Southern Ocean in $m^2$. CONTROL and HOSING simulations are shown in black and red, respectively, and gray lines show the end of the spinup. The start time of HOSING is set to year 0. Solid lines show 10 year running mean of the yearly (lighter coloured) data. The three shaded areas show the 100 year time periods that are discussed further in Sect. 3.2 – 3.4.

The North Atlantic freshwater forcing weakens the AMOC from $21.5\,\mathrm{Sv}$ to $5\,\mathrm{Sv}$ (that we refer to as AMOC collapse) after

100 years in the HOSING experiment (Fig. 1a). This reduction and its impacts are in line with the results of the NaHosMIP models (Jackson et al., 2023; Diamond et al., 2025) in the first 100 years after the AMOC collapse. With the weakening of the AMOC, the surface air temperature in the Northern Hemisphere decreases (Fig. Appendix A1a), particularly over the Atlantic north of $60°\,\mathrm{N}$ (up to $-9\,°\mathrm{C}$), as the northward oceanic heat transport is reduced. In the SH, this results in increased surface air temperature (Fig. A4a) similar to Diamond et al. (2025); Orihuela-Pinto et al. (2022); Pedro et al. (2018); Vellinga and Wood



(2002), as well as in a southward shift of the ITCZ (Fig. A1c), and decreased sea level pressure in the SH (Fig. A1d), especially in the subtropical high regions (Orihuela-Pinto et al., 2022). Due to the collapsed ocean circulation, heat accumulates in the subsurface South Atlantic north of $40°\,S$ (Fig. A1b) as also found in Pedro et al. (2018). This increases the temperature gradient across the ACC, which in combination with strengthened westerly winds leads to an intensification of ACC by approx $20\,Sv$ (Fig. 1c) as suggested by Wu et al. (2021). In the ninth century of the HOSING simulation, there is a climate regime shift, as

the rate of AABW formation increases by more than $10\,Sv$ (Fig. 1b). A similar shift is present in the ACC strength (Fig. 1c) and maximum sea ice extent in the SH abruptly decreases by around $40\,\%$ (Fig. 1d). We analyse these changes in detail in Sect. 3.4.

**Figure 2.** Antarctic Ice Sheet time series: (a) Sea-level-rise potential in m, (b) oceanic temperature forcing in °C as the Antarctic basin mean, (c) total basal mass flux in Gt/year and (d) total calving flux in $\mathrm{Gt\,yr^{-1}}$. The CONTROL and HOSING simulations are shown in black and red, respectively, and gray lines show the end of the spinup. The start time of HOSING is set to year 0. All data are presented with a decadal temporal resolution. The three shaded areas show the 100 year time periods that are discussed further in Sect. 3.2 – 3.4.





Figure 2 shows the transient response of the AIS to an AMOC collapse. The total change in the ice sheet volume is comparably small, as it barely exceeds the range of internal variability during the coupled spinup. Sea-level-rise potential (SLRP) with respect to the CONTROL experiment is decreased by around $2.5\,\mathrm{cm}$ after 1500 years (Fig. 2a), without showing any
abrupt deviations during the whole integration. Because PISM is only exposed to oceanic changes in our configuration, the main drivers of change in the ice sheet are basal melting and calving fluxes. Both fluxes are subject to oceanic temperature and salinity anomalies, which the model extracts from ocean depths between $500$ and $1000\,\mathrm{m}$, depending on the glacial basin. Basal mass and calving fluxes in HOSING are within the range of variability of CONTROL in the first 800 years of the simulation (Fig. 2c, d), because during this period mean subsurface ocean temperature forcing in HOSING is in the range of values in
CONTROL ($-0.85\pm0.4\,^{\circ}\mathrm{C}$) (Fig. 2b). There is less variability in the ocean temperature, explaining the decrease of basal mass variability (Fig. 2b, c). Calving, on average, is $3\%$ larger than in CONTROL, driving the slight decrease in Antarctic mass (Fig. 2a, d). With the shift of AABW formation at around 830 years, the average ocean temperature across all basins decreases to $-1.5\,^{\circ}\mathrm{C}$ in HOSING (Fig. 2b). This drives a subsequent decrease in basal mass flux (mean decrease rate of $160\,\mathrm{Gt\,yr^{-1}}$) (Fig. 2c). In the same period, an increase in calving flux (on average $170\,\mathrm{Gt\,yr^{-1}}$) (Fig. 2d), balances the basal mass decline,
leading to similar rates of SLRP changes in HOSING and CONTROL. The reduction in basal melting leads to growing ice shelves that calve more frequently, which explains the increase in calving fluxes. The average ocean salinity across all basins shows a constant negative (freshening) trend (Fig. A13b) though this trend of decreasing salinity is not reflected in the basal mass fluxes. Overall, in our simulations there is no destabilizing effect of an AMOC shutdown on the WAIS or other parts of the AIS, as hypothesized based on the SO surface warming after an AMOC collapse. Overall, Antarctic ice mass changes
(Fig. 2a) are relatively small in response to the strongly reduced AMOC strength.

To explain this result, the following subsections investigate SO conditions during the HOSING simulation, providing insights into the changes in SO subsurface temperature. We analyse the mean climate states of three time intervals indicated in Figure 1 and 2. The first period (mean of model years 100–200) is chosen to focus on the time interval that previous studies have investigated (Diamond et al., 2025; Pedro et al., 2018). The second period (mean of years 600–700) represents the climate state
before a reduction in maximum Southern Hemisphere (SH) sea ice extent (Fig. 1d) and the onset of deep convection in the Ross Sea. The last time period (mean of years 1400–1500) shows the climate state at the end of our simulation. The climate state of each time period is discussed in the following subsections with a focus on the changes in the SO and adjoint Antarctic ice sheet basins.

## 3.2 Drivers of shorter-term changes in Southern Ocean conditions (years 100–200)

In the first period, the SO is characterized by positive sea-surface temperature (SST) anomalies up to $1.5\,^{\circ}\mathrm{C}$ south of $50\,^{\circ}\mathrm{S}$ (Fig. 3a). This warming leads to a decrease in maximum sea ice extent and decreasing sea-ice thickness by $5$ to $10\,\mathrm{cm}$ in all coastal regions around Antarctica (Fig. 1d, A4d). Sea-ice melt dominates the change in total surface freshwater flux over changes in the precipitation-minus-evaporation balance (Fig. A8a, d) which can regionally decrease surface density. This is outweighed, however, by positive sea-surface salinity (SSS) anomalies originating in the South Atlantic (Fig. 3b), which diffuse
across the ACC leading to a net increase in surface density in much of the SO (Fig. 3c), particularly the Weddell Sea region. In





CONTROL, the Weddell Sea is the only SO region where convection occurs. This vertical mixing changes little in HOSING (Fig. 3e), shifting southwards due to reduced sea ice and more negative wind stress curl (Figures 3f, A4d).

The atmospheric warming (Fig. A4a) south of $65°$ S leads, on average, to an ocean warming from the surface down to around $250\,\mathrm{m}$ (Fig. A10a-c). However, temperature anomalies in the subsurface (500-1000 m) in this region are negative (Fig. 3d)

between years 100 and 200. This cooling originates in the North Atlantic, as residual NADW transports the cooling signal of the collapsed AMOC to the SH. The inflowing NADW is upwelled mainly in the Atlantic sector and reaches the waters close to the Antarctic continent approximately in model year 100 (Fig. 4b). The ACC transports this signal along the coast, spreading it to all sectors of the SO (Fig. A10a-c). In regions without convection, we find that the cooling strengthens with time, and inflowing water masses also drive a freshening trend (decrease of $0.1\,\%$ per year) in the same depths (Fig. 4d). These changing

subsurface conditions are decisive for the evolution of the AIS. The temporal evolution of basal mass flux is strongly aligned with the mean subsurface ocean temperatures (between $500$ and $1000\,\mathrm{m}$ depth), which decrease by around $0.15\,°\mathrm{C}$ (Fig. 2b). Therefore, between years 100 and 200, averaged changes of basal mass flux show a decrease up to $100\,\mathrm{Gt\,yr^{-1}}$ after 200 years in HOSING (Fig. 2c). Resulting AIS thickness changes are limited to the coastal regions and mostly smaller than $5\,\mathrm{m}$ (Fig. 3d). Despite the decrease of basal melt in most regions, there is a peak of mean basal mass flux in the Atlantic sector around year

90 of our simulation (Fig. A11, basin 1). It results due to several years with less convection in which subsurface temperature anomalies in the Weddell Sea are positive (maximum positive anomaly of $0.4\,°\mathrm{C}$ between year 80 and 130) (Fig. 4b). The reduced vertical mixing stops the heat release from subsurface waters to the atmosphere (Fig. A5d), driving a briefly increase of melt rate up to $200\,\mathrm{Gt\,yr^{-1}}$ that is reflected in decreasing AIS thickness of the Filchner-Ronne ice shelf. These thickness changes dominate the 100-year mean AIS thickness anomaly (Fig. 3d). Local ice-sheet changes are therefore directly linked to

the changing frequency of convection in the Weddell Sea, which impacts interior temperature and salinity changes of coastal ocean waters. Similar fluctuations also exist in the spinup and CONTROL and are attributable to the convective activity in the Weddell Sea.





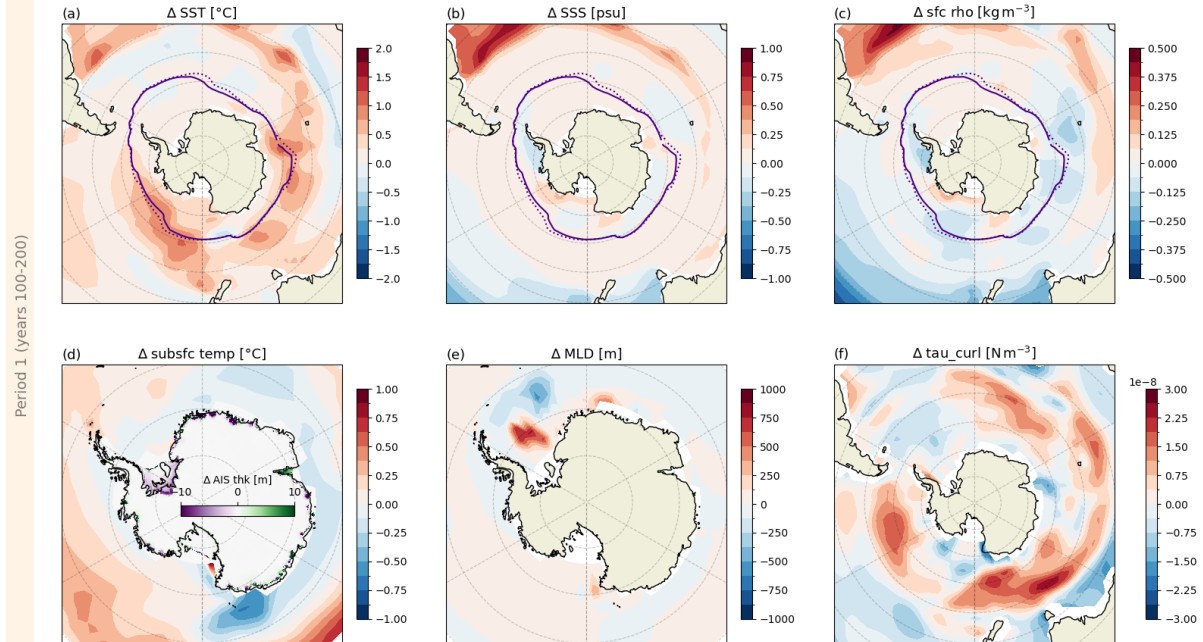

**Figure 3.** Southern Ocean anomalies with respect to CONTROL of (a) sea surface temperature in °C, (b) sea surface salinity in psu, (c) sea surface density in $\mathrm{kg\,m^{-3}}$, (d) averaged ocean subsurface temperatures between $500$ and $1000\,\mathrm{m}$ in °C and land ice thickness on the AIS in m, (e) maximum mixed layer depth in m and (f) wind stress curl in $\mathrm{N\,m^{-3}}$. Each panel shows anomalies with respect to CONTROL as a 100 year average for the first time period indicated in Fig. 1. Purple contours in the top row show the maximum sea-ice extent where concentration is larger than $15\,\%$ per grid cell for CONTROL (dotted line) and HOSING (solid line). Latitude graticules are plotted with a $10°$ grid spacing.

## 3.3 Reduced Southern Ocean convection leading to subsurface heat accumulation (years 600–700)

In the intermediate part of our simulation (years 600–700), atmospheric surface-air temperatures over and close to Antarctica show a cooling signal in most regions south of $60°\,\mathrm{S}$ other than over Prydz Bay in East Antarctica (Fig. A4b). The maximum cooling in this time period is $-3\,°\mathrm{C}$ over parts of the Weddell Sea and over the Amundsen and the Ross Sea. Consequently also SSTs in these regions show negative anomalies (Fig. 5a). After $\approx 250$ model years, sea ice in the SH is able to regrow and its maximum extent exceeds that of the CONTROL run (see contour lines in Fig. 5a–c). Sea-ice thickness increases in the Weddell and Ross Sea region (Fig. A4e) by $5$ to $20\,\mathrm{cm}$ and variability of the averaged maximum sea-ice extent decreases compared to CONTROL (Fig. 1d). The wind stress curl at the ocean surface intensifies along the maximum sea-ice edge, especially in the Bellingshausen and Amundsen Sea (Fig. 5f). Thereby, fresher circumpolar deep water is upwelled, conveying the low salinity signal of the North Atlantic hosing (Fig. 4c, d). We therefore see negative SSS anomalies (Fig. 5b), despite less freshwater flux due to sea-ice melt south of $70°\,\mathrm{S}$ (Fig. A8e), and surface densities south of $60°\,\mathrm{S}$ decrease on average by $0.25\,\mathrm{kg\,m^{-3}}$ (Fig. 5c) in all sectors except for the eastern part of the Pacific sector and latitudes north of $70°\,\mathrm{S}$ in the Pacific. Positive sea-surface





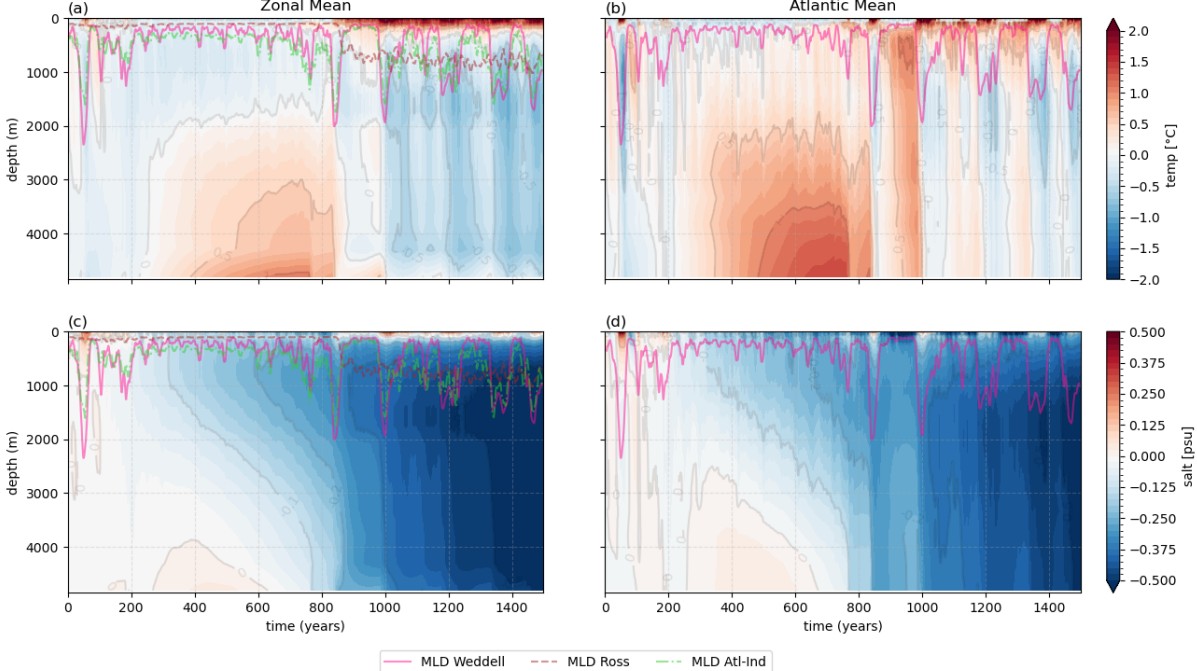

**Figure 4.** Hovmöller plots of (a), (b) temperature and (c), (d) salinity anomalies in °C / psu with respect to a 500-year mean of the CONTROL in the Southern Hemisphere, south of $65°$ S. Subplots (a) and (c) show averages over all longitudes, (b) and (d) averages over the Atlantic Sector. Overlaid time series show the maximum mixed layer depth (MLD) averaged over the Weddell Sea (lon=$[60°$ W-$0°$ W], lat$<-65°$ S) (pink line), the Ross Sea (lon=$[150°$ E-$150°$ W], lat$<-65°$ S) (brown, dashed line) and the east of the Weddell Sea (Atl-Ind) (lon=$[0°$ E-$60°$ E], lat$<-65°$ S) (green, dash-dotted line).

salinity anomalies (up to $+0.5$ psu) in the eastern Bellingshausen Sea (Fig. 5b) are probably transported there from the South Atlantic, as the Ross Sea gyre is strengthened over time (Fig. A7e) and increases the water transport from the ACC towards the margin of the Antarctic continent (Wang et al., 2024).

Decreased densities of upwelled waters increase surface stratification, leading to a strong decrease in vertical mixing (maximum mixed layer depth (MLD) reduction of around $1250$m) in the Weddell Sea (Fig. 5e). Therefore, variability of AABW

formation (Fig. 1b) decreases in this time period. The reduced convection drives negative SST anomalies in the Atlantic sector south of $60°$ S (Fig. 5a) and subsurface temperatures in the Atlantic sector increase by a maximum of $0.5$ °C at depths of $500$–$1000$ m (Fig. 5d), as sensible and latent heat loss to the atmosphere is reduced (Fig. A5e). As a result, basal mass fluxes slightly increase in this region compared to CONTROL (Fig. A11, basin 1), however AIS thickness anomalies have the same magnitude as in the previous time period (see Sect. 3.2), i.e. ice loss does not accelerate (Fig. 3d). With the reduction in Weddell

Sea convection, heat begins accumulating in the deep SO below $2000$ m in all ocean basins (Fig. 4a, b, A9a, b). Salinity values below $2000$ m show increasing negative anomalies, which originate from subsurface depths and over time diffuse to the deep SO (Fig. 4c).



In all other ocean sectors, the density of intermediate waters continue to decrease, driven by the freshening of upwelled NADW in the SO that is transported around the Antarctic continent by the ACC, as discussed for the previous time period in Sect. 3.2. Therefore, in the Indian and Pacific sector we find negative subsurface temperature anomalies (up to $-1\,^{\circ}\mathrm{C}$) along the coast (Fig. 5d), as well as decreasing salinity in subsurface depths in all regions south of $50^{\circ}\,\mathrm{S}$, up to $-0.45\,\mathrm{psu}$ (Fig. A3e). These changes in subsurface ocean forcing (temperature and salinity) between years 200 and 800 therefore show the continuation of the trends seen in the previous time period. Consequently, the response of the AIS is very similar, showing an averaged change of basal mass flux decrease of $100\,\mathrm{Gt\,yr^{-1}}$ with respect to CONTROL (Fig. 2c) and an increase in calving flux of the same rate (Fig. 2d). Both signals are most evident in ice-sheet basins in the Ross Sea region as well as basins close to and east of the Amery ice shelf (Fig. A11), where also AIS thickness anomalies are positive (Fig. 5d).

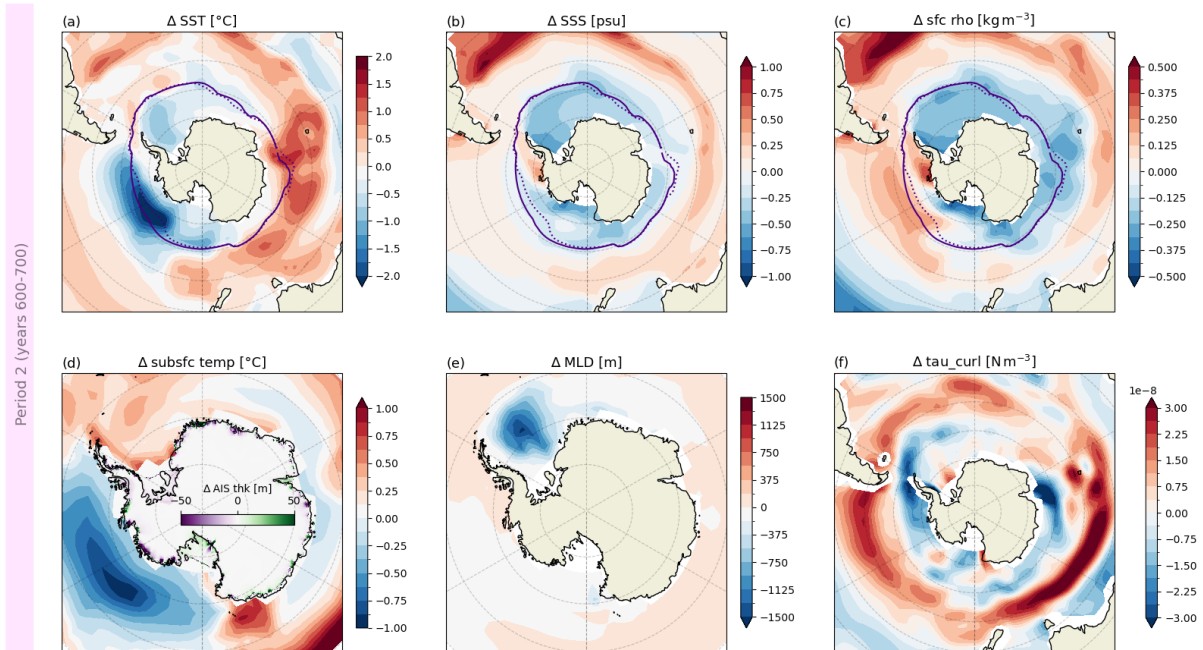

**Figure 5.** Southern ocean anomalies of (a) sea surface temperature in $^{\circ}\mathrm{C}$, (b) sea surface salinity in psu, (c) sea surface density in $\mathrm{kg\,m^{-3}}$, (d) averaged ocean subsurface temperatures between $500$ and $1000\,\mathrm{m}$ in $^{\circ}\mathrm{C}$ and land ice thickness on the AIS in m, (e) maximum mixed layer depth in m and (f) wind stress curl in $\mathrm{N\,m^{-3}}$. Each panel shows anomalies with respect to CONTROL as a 100 year average for the second time range indicated in Fig. 1. Purple contours in the top row show the maximum sea ice extent where concentration is larger than $15\,\%$ per grid cell for CONTROL (dotted line) and HOSING (solid line). Latitude graticules are plotted with a $10^{\circ}$ grid spacing.

### 3.4 Southern Ocean heat release by increased convection (years 1400–1500)

From year 700 onwards, vertical mixing in the Weddell Sea progressively increases, down to $2000\,\mathrm{m}$ depth (Fig. 4a). Convection in the Ross Sea starts to ventilate the deep SO (Fig. 6e) from year 830 onwards, mixing water columns in this region



continuously every year (Fig. 4b). Convection starts due to the growing water column instability driven by the accumulation of heat in the deep SO in previous centuries (Fig. 4a) as well as decreased salinity of subsurface waters (Fig. 4c). As a result, AABW formation in HOSING increases by $15\,\mathrm{Sv}$ with respect to CONTROL at the end of the simulation (Fig. 1b), exhibiting an anti-phased behaviour of NADW and AABW formation (Barbante et al., 2006; Peltier and Vettoretti, 2014; Skinner et al., 2020; Willeit et al., 2025). Another contributor to the destabilisation of the SO water column might be changes in temperature

of NADW, which is upwelled in the SO. As in the study by Pedro et al. (2018), heat accumulates in the South Atlantic, where over time it gradually diffuses to the deep ocean (see Video Supplement S1). This diffusion is a continuous process during the whole simulation, warming over time the NADW that flows to the SH. Consequently, the negative subsurface temperature anomalies in the SO diminish over time and switch to positive anomalies around year 800 (see Video S1). Around one century after the convection onset in the Ross Sea, water columns also reach instability in other SO sectors and therefore convection

starts in coastal regions eastward from the Weddell Sea to around $70°\,\mathrm{E}$ (the ocean in front of the Amery Ice Shelf) (Fig. 6e).

Along with the strengthening of convection in the SO there is an increase of global mean SST (Fig. A13a) driven by the release of subsurface heat (Fig. A5f) due to vertical mixing in the SO (Fig. 6e). The SST increase around the Antarctic continent is around 4 times higher in the climate state after AABW onset (Fig. 6a) than after the atmospheric warming due to the AMOC collapse (Fig. 3a). In line with this warming, there is a shift in SO sea-ice extent, decreasing to around $55\,\%$ of the extent in

CONTROL (Fig. 1d). Sea ice is thinning in all regions around Antarctica by up to $50\,\mathrm{cm}$ (Fig. A4f).

Furthermore, the increase of deep convection leads to a cooling of Antarctic Bottom waters, averaged along the continental slope of Antarctica up to $\approx 65°\,\mathrm{S}$, of up to $1.2\,°\mathrm{C}$ with respect to CONTROL, except for the surface layer above $\approx 500\,\mathrm{m}$ (Fig. 4a). Due to the climate regime shift in the SO in the last time period the Ross Sea gyre shifts its location again, so that with respect to CONTROL it expands to the east Indian sector in front of Wilkes Land (Fig. A7f). Thereby, cool water masses

are transported westward from the Ross Sea. As a consequence, subsurface temperatures in the East Antarctic also decrease by up to $2\,°\mathrm{C}$ (Fig. 6d). In the adjacent ice-sheet basins, the basal mass flux reduces to values close to zero. Especially in the East Antarctic, the calving rates increase by up to $100\,\mathrm{Gt\,yr^{-1}}$ compared to CONTROL as ice shelves grow larger with less basal melt. In the Weddell Sea subsurface temperatures partly increase along the coast (Fig. 6d), which can be explained by shallower mixed layer depth in the eastern Weddell Sea (Fig. 6e). Reduced heat loss to the atmosphere (Fig. A5f) leads to less

pronounced SST warming compared to most other regions around Antarctica (Fig. 6a) and temperature anomalies at 500 to $1000\,\mathrm{m}$ depth are positive $(0.05 - 0.3\,°\mathrm{C})$ along the coastal margin (Fig. 6d). In basins located nearby the Weddell Sea, basal melt increases in years of positive temperature forcing, i.e. only for certain years when no convection site opens up. Basins with ice shelves in Bellingshausen Sea show no significant change in basal mass or calving fluxes (Fig. A11, A12).





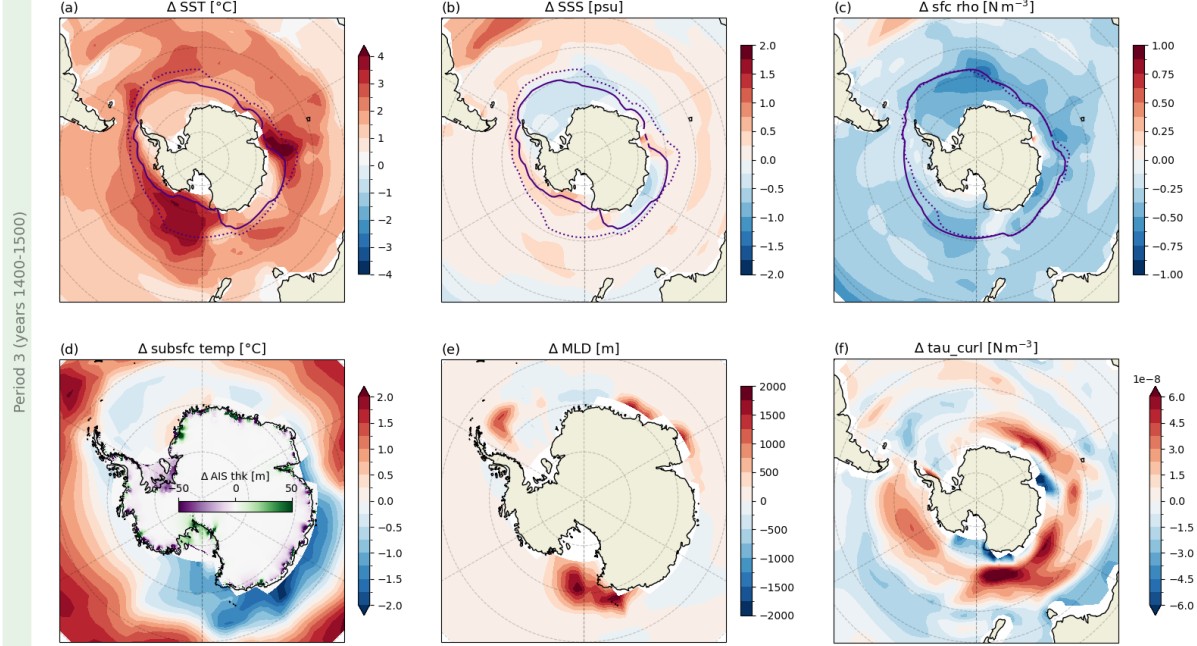

**Figure 6.** Southern ocean anomalies of (a) sea surface temperature in °C, (b) sea surface salinity in psu, (c) sea surface density in $\mathrm{kg\,m^{-3}}$, (d) averaged ocean subsurface temperatures between $500$ and $1000\,\mathrm{m}$ in °C and land ice thickness on the AIS in m, (e) maximum mixed layer depth in m and (f) wind stress curl in $\mathrm{N\,m^{-3}}$. Each panel shows anomalies with respect to CONTROL as a 100 year average for the third time range indicated in Fig. 1. Purple contours in the top row show the maximum sea ice extent where concentration is larger than $15\,\%$ per grid cell for CONTROL (dotted line) and HOSING (solid line). Latitude graticules are plotted with a $10°$ grid spacing.

## 4 Discussion

The novelty of our study is to simulate the response of Antarctica to an AMOC collapse directly in an interactively coupled climate and ice-sheet model. We have extended the results of previous freshwater hosing model studies (e.g. Diamond et al., 2025; Orihuela-Pinto et al., 2022; Stouffer et al., 2006), which are mainly focused on atmospheric feedbacks in the Northern and Southern Hemisphere. To capture multi-centennial responses of the AIS, we integrate the model for 1500 years. Even though we could only run one realisation, the simulation length captures possible regime shifts due to internal variability.

Contrary to our hypothesis and previous assumptions (see review by Wunderling et al., 2023), the WAIS stays stable during a period with collapsed AMOC, driven primarily by persistently cold temperatures at the depths of the ice shelf cavities. Climate impacts in the first 200 years after HOSING are in good agreement with previous model studies (Diamond et al., 2025; Jackson et al., 2023), which use state-of-the-art CMIP model configurations. In particular, in the SO we find a shift of convection in the Weddell Sea in the first century after the AMOC collapse, which partly warms temperatures in the depth range $500$ to $1000\,\mathrm{m}$,

similar to the findings of Yeung et al. (2024) for different (last interglacial) boundary conditions. This warming lasts for around





one decade and leads to an increase in basal melting which, however, remains within the range of natural variability in the control simulation (CONTROL) and is confined to the Weddell Sea. In other regions, Antarctic ice mass does not change.

The AIS response in HOSING slightly shifts after 830 years, as basal melt decreases due to cooling at the depths of ice-shelf cavity inflow. This cooling is caused by an increase in vertical mixing in the SO, which corresponds to the multi-centennial
ocean response to the strong AMOC weakening, showing an increase of AABW formation as suggested by Broecker (1998); Skinner et al. (2020); Willeit et al. (2025). Different studies propose that the oceanic connections between NADW and AABW strength has a different time-scale than the signal propagation of changes by the atmosphere. For example, Swingedouw et al. (2009) distinguish the bipolar climate seesaw (BCS) from the bipolar ocean seesaw (BOS) in the context of a SO freshwater release experiment or Skinner et al. (2020) find that during the Heinrich Stadial 4 enhanced SO convection resulted due to
the ventilation seesaw. To the best of our knowledge, our AMOC collapse study is the first one showing an abrupt increase of AABW strength, possibly because model integration times of previous studies were too short. However, Berdahl et al. (2024) did a hosing experiment for 4000 years using CESM2 and discuss a similar cooling of subsurface temperatures during an AMOC collapse. In their study, they do not analyse how ocean convection sites change in the model nor address changes in AABW formation. They explain the subsurface cooling in their simulation by an increase in wind stresses over the SO. Other
studies (Rind et al., 2001; Kageyama et al., 2010) with similar setups also show cooling subsurface temperatures in the SO after maximum 500 years of hosing, yet do not address the possible consequences for the AIS. Output of the NaHosMIP models (see Supplemental Material of Diamond et al., 2025) shows that changes in the SO subsurface on the centennial timescale vary dramatically between different models (cooling in CESM2, warming in HadGEM3-GC3.1-LL), emphasising that the location, vertical extent and intensity of open ocean convection in the SO are crucial for changes in subsurface water properties, which
have direct impacts on the AIS. Even though it takes many centuries for our model to reach an ocean state with strengthened AABW formation in the SO, we note that the impact of this shift is significantly stronger in the Southern Hemisphere than the one of the BCS, which supports findings of Pedro et al. (2016) and Skinner et al. (2020). However, as our study is the first hosing simulation using a coupled climate–ice-sheet model, our results can be unique to our model and configuration, especially the timing of convection onset. Therefore, it is crucial that others repeat this experiment using coupled models, to
be able to draw more general conclusions about the interaction between the AMOC and the AIS.

To be able to compare the study easily with other model runs, we chose to design the freshwater hosing experiment based on the NaHosMIP protocol (Jackson and Wood, 2018). We note that due to the artificial freshwater flux in the Northern Hemisphere, which we apply without compensation, salinity in the ocean decreases constantly throughout the simulation. Stocker et al. (2007) shows, that on multi-decadal time-scales salt compensation in AMOC collapse experiments does not
contribute significantly to the temperature response in the SH. Compensating for the freshwater flux globally changes SSS in the SO and thereby has (in our model) direct implications for convection by affecting SO stratification. However, without applying any compensation of the artificial freshwater forcing, the total sea level in our model rises to $\approx 38.5\,\mathrm{m}$ after the 1500 model years. This change translates to $0.26\,\mathrm{m}$ sea level height change each coupling time step (one decade), which we assume to have no significant influence on the SO structure around Antarctica. The applied freshwater flux in the Arctic of $0.3\,\mathrm{Sv}$ is
around 7 times higher than recent estimates of Greenland freshwater forcing (Wouters and Sasgen, 2022), however is needed



to keep the AMOC in a collapsed state. Like all hosing experiments, our setup is therefore strictly idealised, which nonetheless is an useful approach to advance our understanding of the impacts of an AMOC shutdown on other components of the Earth system.

Our analysis focuses mainly on the significant temperature changes at intermediate depths of the SO, as the trend of decreas-
ing salinity is not reflected in the AIS basal mass fluxes. This can be explained, in part, by the temperature sensitivity of PICO (Kreuzer et al., 2025), which is discussed with the PICO limitations below. However, the decreasing salinity trend is crucial to the changes in AABW formation in HOSING from model year 850 onwards. With the current setup, we cannot distinguish between freshening of circumpolar deep water due to the artificial freshwater hosing or due to less salt mixing into the deep ocean after a collapse of the AMOC. Future work is needed, to understand the water export from the hosing region and its
impacts in the SO in more detail.

The main motivation of choosing a comparatively low resolution of our climate model is to allow the relatively long integra-
tion time of 1500 years plus spinups, though this inevitably leads to shortcomings in the representation of important processes. AABW in the SO is exclusively formed by open ocean convection rather than the export of dense shelf water. Vertical mixing events result as a consequence of weak stratification in the Weddell Sea and Ross Sea (Galbraith et al., 2011). Similar open
ocean convection sites occur in 80% of CMIP6 models (Heuzé, 2021), and are a common weakness in global climate models running at non-eddy-resolving resolutions. In our model setup, deep ocean convection has a large impact on coastal waters at depths where we extract temperature and salinity values with which to force PICO-PISM. The limited resolution of our ocean model lacks features such as the Antarctic slope current that might otherwise block direct signal propagation. The AIS response after ca. 800 years might be distorted by the open-ocean convection regime change. Furthermore, the representation of transport
via mesoscale eddies, which plays an important role in heat transport across the ACC, is simplified by the Gent-McWilliams parameterisation (Gent and McWilliams, 1990). As investigated by Pedro et al. (2018), the timing of heat propagation in the SO is probably slower than in eddy resolving models, which would affect the timing of the changes we observe.

The coupling of climate and ice-sheet model (Kreuzer et al., 2021) is based on the sub-module PICO of PISM, which parameterizes the overturning ocean circulation in ice shelf cavities, depending on temperature and salinity of inflowing water
masses. PICO bridges the gap between even simpler basal melt parameterisations and high-resolution cavity-resolving ocean models (Burgard et al., 2022). It does not capture all fine-grained details of horizontal melt patterns, but has been proven to compute realistic bulk melt rates and melt-rate sensitivities, locally and on a circum-Antarctic scale, for historic and future scenarios (Reese et al., 2018, 2023).

Using PICO enables the co-evolving simulation of the climate and the AIS on millennial timescales by capturing changes of
ice fluxes in dependence on the prevailing ocean forcing, and adding resulting meltwater and heat fluxes at realistic depths back into the ocean model. Hence, the model framework captures the expected feedback of increasing basal melt rates in the ice sheet due to increased subsurface ocean temperatures (Paolo et al., 2015) and vice versa. As pointed out by Kreuzer et al. (2025), PICO is more sensitive to temperature forcing than to salinity forcing as melt rates are estimated depending on the equation of state (eq. 8 in Reese et al., 2018). Salinity forcing in our results shows a decline, that would force melt rates to decrease,
as the freezing point is at higher temperatures compared to saltier water. This change would therefore impact basal melt in the





same way as temperature does, so that the temperature sensitivity of PICO does not distort the AIS response. The reduction in basal melt seen in most regions around Antarctica leads to less (negative) heat flux extracted from the ocean to account for phase change energy and therefore a small increase in temperature in the depth where basal melt fluxes are inserted into ocean column. This pattern cannot be observed in the centennial mean climatology, as it is dominated by temperature changes

resulting from changes in circumpolar deep water and vertical mixing. Due to the limited spatial and temporal resolution of this study, it is unclear whether strong seasonality or unresolved mesoscale processes would cause a destabilising trend of the ice sheet equilibrium state. For example, the coupling framework works with decadal anomalies, which smooths out possible extreme values that could cause extremer signals in the AIS/ocean. Nonetheless, we do not expect high impacts due to this limitation, as the AIS usually reacts rather slow (compared to other Earth system components) and no long-term trends in our

model indicate that this would be the case.

The sensitivity of the PISM spinup to changes in ocean properties, which can differ greatly depending on the ice-sheet spinup procedure, is not investigated here. In Garbe et al. (2020), a collapse of the WAIS is discussed as a response to increased ocean temperature forcing. This collapse results from marine ice sheet instabilities in regions with retrograde sloping bedrock, even though it is argued that the setup is not in line with observations (Mouginot et al., 2014; Joughin et al., 2014). Therefore they

suggest that especially ice shelves in the Amundsen Sea are more sensitive to an increased ocean forcing than in their study. As our study adopted the spinup state from Garbe et al. (2020), the PISM configuration used here might also be too stable compared to the present-day AIS state.

The model setup used for this study has no coupling between PISM and the AM2 atmosphere. Therefore, external boundary forcing from the atmosphere to the ice sheet (surface air temperature and precipitation) remains constant throughout the

simulation period. However, PISM uses of an internal parameterisation of temperature adjustment with height (see Sect. 2.1), which enables a representation of the melt-elevation feedback. Given the fact, that under pre-industrial and present-day climate conditions most changes in Antarctica are due to interactions at the ice-ocean interface, a missing ice-atmosphere coupling might be reasonable. However, since air temperatures over the ice sheet and shelves increase between $0.5\,°C$ and $5\,°C$ due to the AMOC collapse (Fig. A4c) and also precipitation rates increase (not shown), an additional coupling at the ice-atmosphere

interface probably could increase changes in ice mass loss. Especially in coastal regions, changes of surface mass flux are not always negligible. For example, a study by Lai et al. (2020) found that surface fracturing caused by meltwater ponds at the surface of the AIS increases the risk of further ice loss in Antarctica. Furthermore, increased ice sheet surface melting can lead to reduced buttressing or cause a stronger calving flux which in turn can destabilise the ice sheet regionally (Noël et al., 2023; Gilbert and Kittel, 2021). Nevertheless, on centennial time-scales atmospheric warming is expected to increase surface

mass fluxes much slower than changes in the ocean, as air temperatures over vast areas of the AIS are far below the melting point. Though we have, for the first time, simulated the response of an AMOC shutdown with an interactive AIS, the limited interaction between ocean and ice sheet highlights the need for more integrated coupling, including atmospheric interactions, in future work.

Although the model setup used in this study does not capture all processes, in particular at small spatial scales, our analysis

helps to understand the relevant mechanisms in coarse resolution models. Furthermore, our results highlight that further studies



are crucial to better understand the effects of an AMOC shutdown on the Antarctic ice sheet. In particular, the non-linearity of the system and the complex behaviour of the SO could prevent the assessment of tipping point interactions based on a single variable that describes the response of the surface ocean (Wunderling et al., 2023). Given the limitations of our approach discussed above, more studies are needed running similar experiments with different coupled model setups. Finally, it would be

valuable to investigate the interaction between the AMOC and the AIS in a more realistic scenario, e.g. running the experiments under high emission scenarios or using a more realistic freshwater forcing amplitude that mimics Greenland ice-sheet melting.

## 5   Conclusions

Our study presents a simplified interactively coupled climate–ice sheet model to investigate the impact of an AMOC collapse on AIS fluxes. Simulating a 1500 year freshwater hosing experiment, we find no destabilisation of the AIS via the ice-ocean

interface. Due to changes in SO convection, induced by imported circumpolar deep water, subsurface SO anomalies are cooling and freshening, which results in reduced basal melting. This change is balanced by calving fluxes as larger ice shelves calve more often. Locally, in our model, changes in the subsurface SO are primarily driven by changes in deep convection. Therefore, the Weddell Sea, where convection happens in a control run, is the only region where temporarily positive temperature anomalies increase ice mass loss from the AIS. Our results indicate that warming SST after an AMOC collapse in the SO

are not necessarily sufficient to driven AIS changes, yet anomalies of the ocean waters in depth of ice-shelf cavities might be crucial. Although our methodology has some simplifications, it is the first hosing experiment using a climate–ice sheet model to investigate the millennial response of the AIS. Nevertheless, higher resolutions would be valuable to verify our results, in particular in the SO, where eddy fluxes and dense shelf water formation on the continental shelf are main drivers of AABW formation under present-day climate conditions. Additionally, in a scenario of a collapsed AMOC ice-atmosphere interactions

in Antarctica could also become more important, leading to increased mass loss or potential instabilities of the AIS.

*Code and data availability.*   All code used in this study is published on Zenodo: The version of the CM2Mc climate model (Kreuzer, 2025a), the version of the ice-sheet model PISM (Khrulev et al., 2025), and the framework to couple CM2Mc and PISM for this work (Kreuzer, 2025b). Also, the model output data used in the study are available at Zenodo (Höse, 2025).

*Video supplement.*   Supplementary video has been uploaded to https://av.tib.eu/, but not published yet at date of manuscript submission.

**Appendix A**



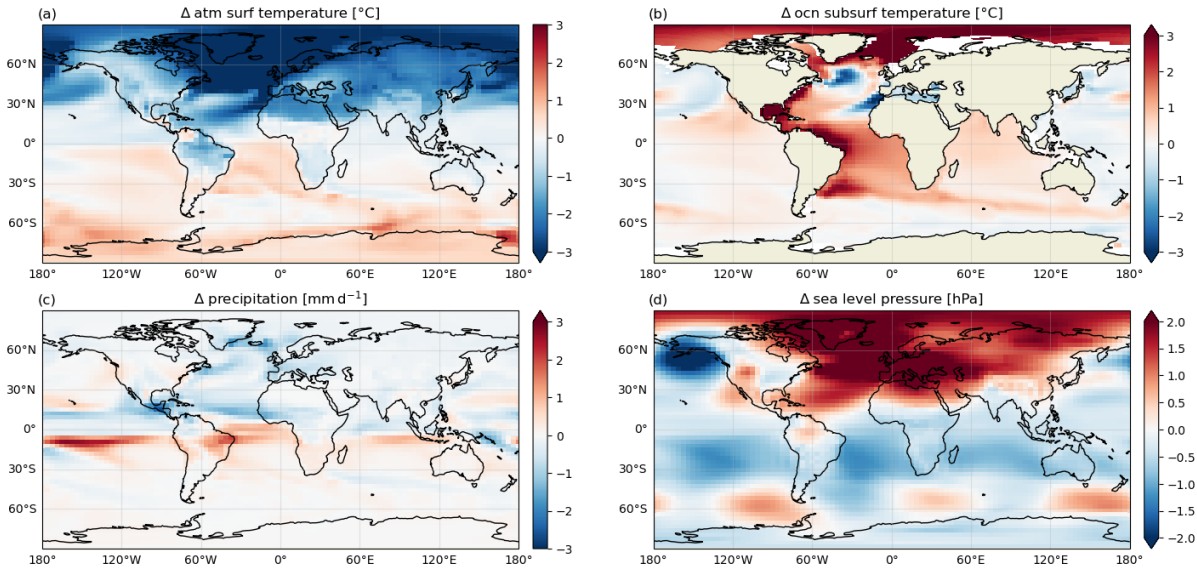

**Figure A1.** Anomalies of (a) sea surface temperature in °C, (b) subsurface ocean temperature (mean between 500 and 1000 m) in °C, (c) precipitation rate in mm d$^{-1}$ and (d) sea level pressure in hPa averaged over the first time period (i.e. mean of years 100–200).




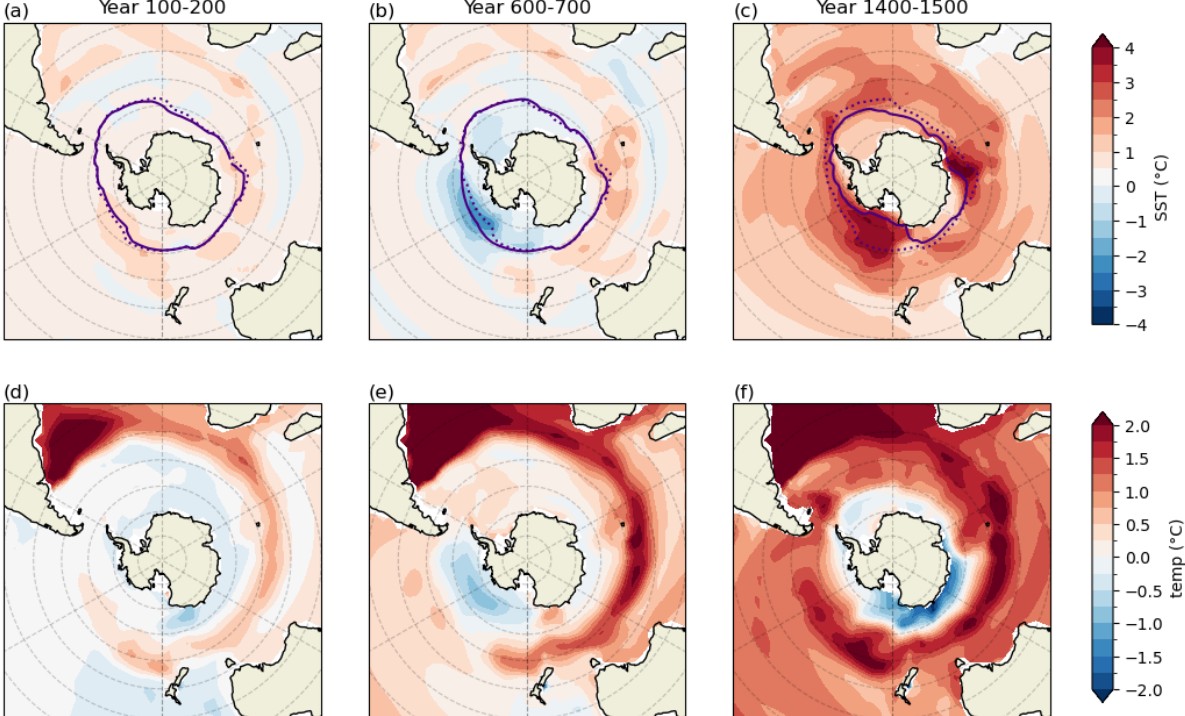

**Figure A2.** Southern ocean anomalies of (a)-(c) sea surface temperature in °C and (d)-(f) averaged subsurface temperatures between 500 and 1000 m in °C. Each column shows a 100 year average for time ranges indicated in Fig. 1. Purple contours in row one show the time averaged maximum sea ice extent where concentration is larger than 15% per grid cell for CONTROL (dotted line) and HOSING (solid line). Latitude graticules are plotted with a 10° grid spacing. Displayed maps are similar to subplots a and d in Figures 3, 5 and 6, but shown here in unified colormap ranges for easy comparison.



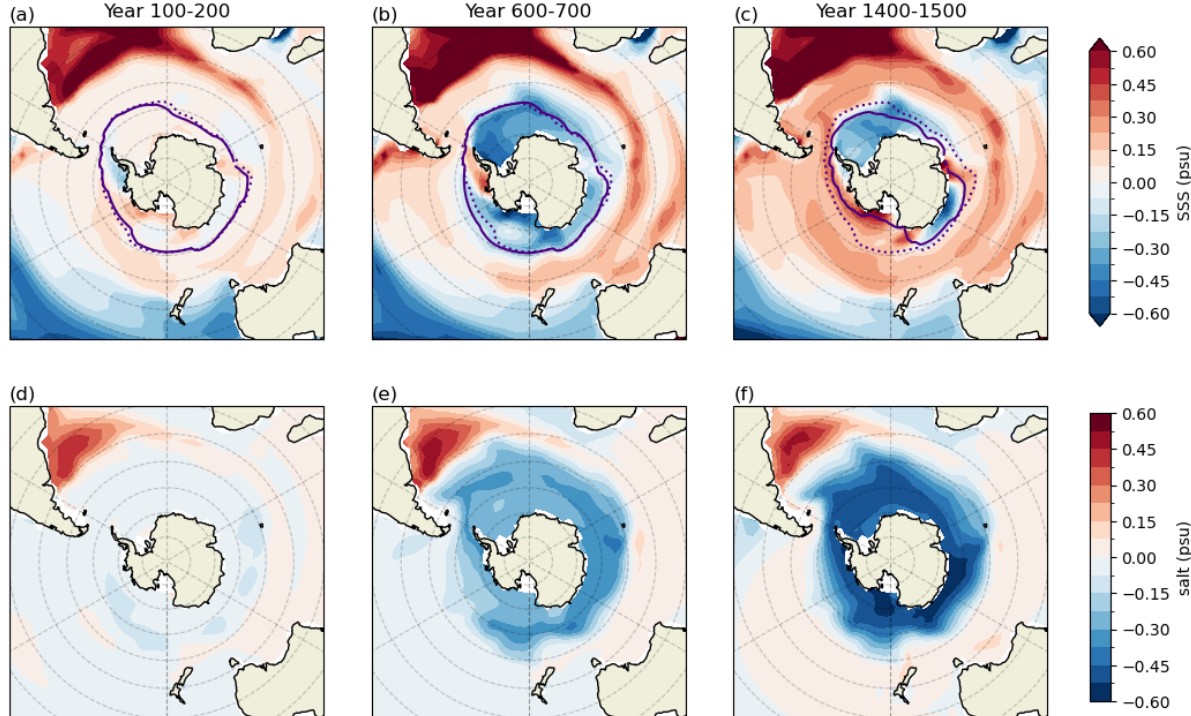

**Figure A3.** Southern ocean anomalies of (a)-(c) sea surface salinity in psu and (d)-(f) averaged subsurface salinities between 500 and 1000 m in psu. Each column shows a 100 year average for time ranges indicated in Fig. 1. Purple contours in row one show the time averaged maximum sea ice extent where concentration is larger than 15% per grid cell for CONTROL (dotted line) and HOSING (solid line). Latitude graticules are plotted with a 10° grid spacing. Displayed maps are similar to subplots b and e in Figures 3, 5 and 6, but shown here in unified colormap ranges for easy comparison.





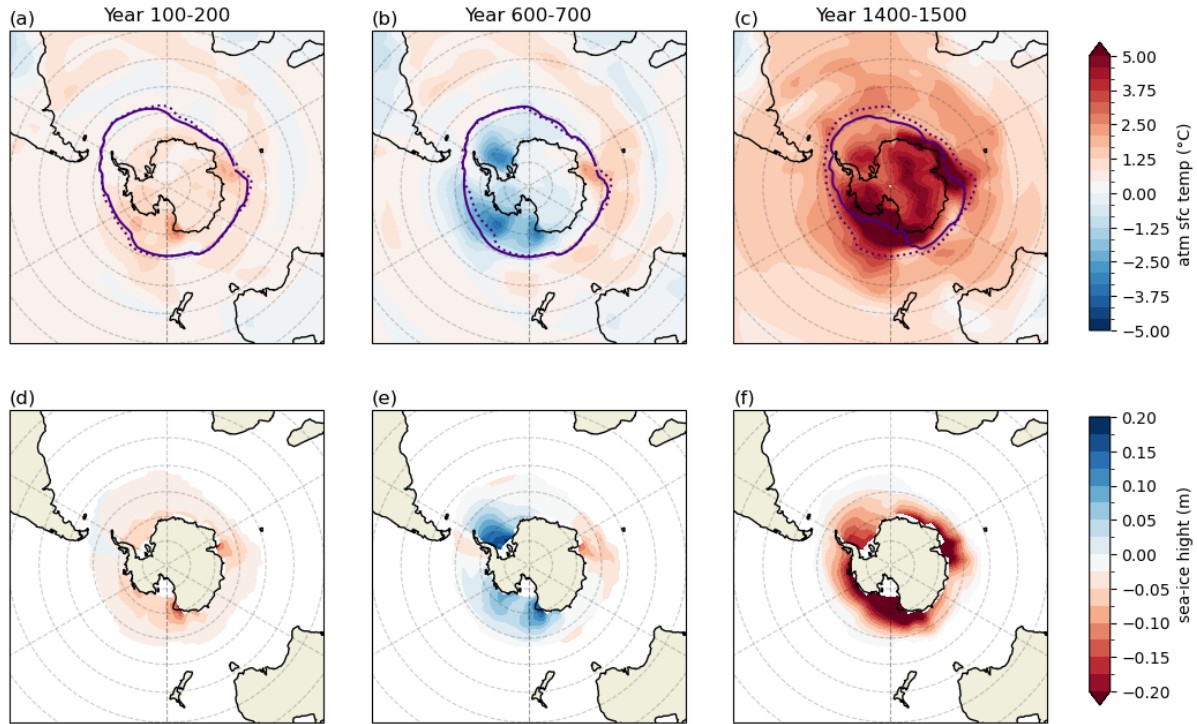

**Figure A4.** Anomalies of (a)-(c) atmospheric surface air temperature in °C and (d)-(f) sea-ice thickness in m. Each column shows a 100 year average for time ranges indicated in Fig. 1. Purple contours in (a)-(c) show the time averaged maximum sea ice extent where concentration is larger than 15% per grid cell for CONTROL (dotted line) and HOSING (solid line). Latitude graticules are plotted with a 10° grid spacing.





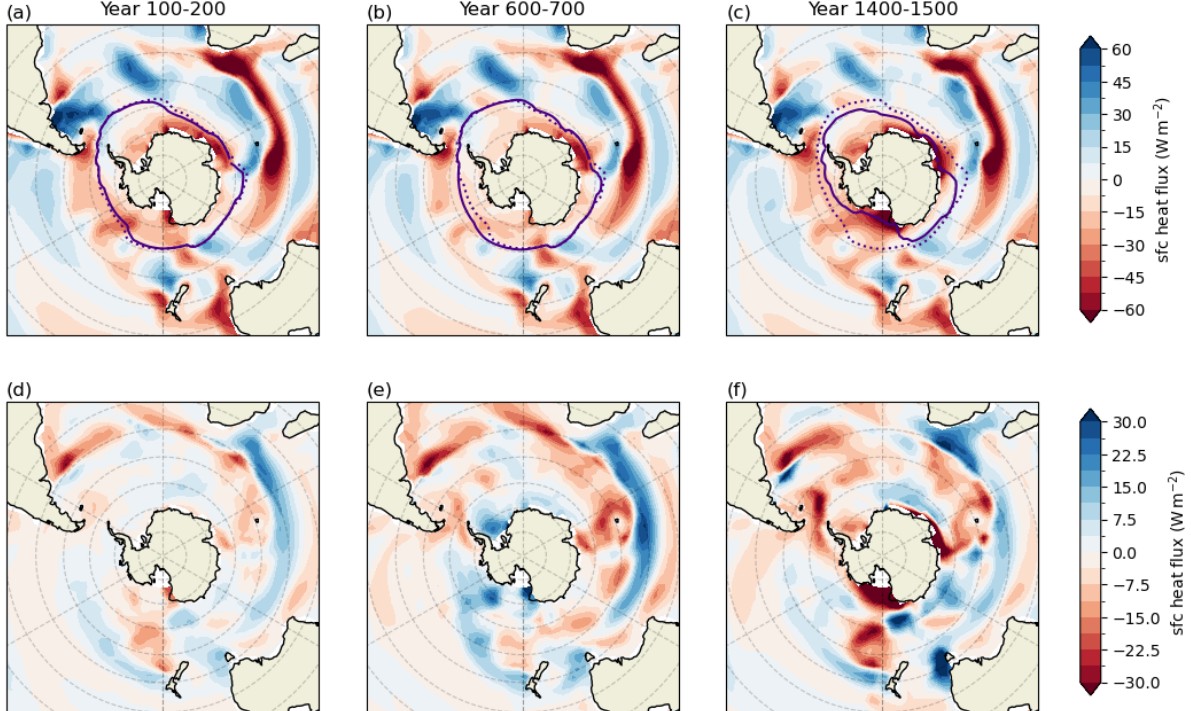

**Figure A5.** Southern ocean (a)-(c) HOSING surface heat flux in $\mathrm{W\,m^{-2}}$ and (d)-(f) anomalies of surface heat flux in $\mathrm{W\,m^{-2}}$. Each column shows a 100 year average for time ranges indicated in Fig. 1. Purple contours in row one show the time averaged maximum sea ice extent where concentration is larger than 15% per grid cell for CONTROL (dotted line) and HOSING (solid line). Latitude graticules are plotted with a $10°$ grid spacing.




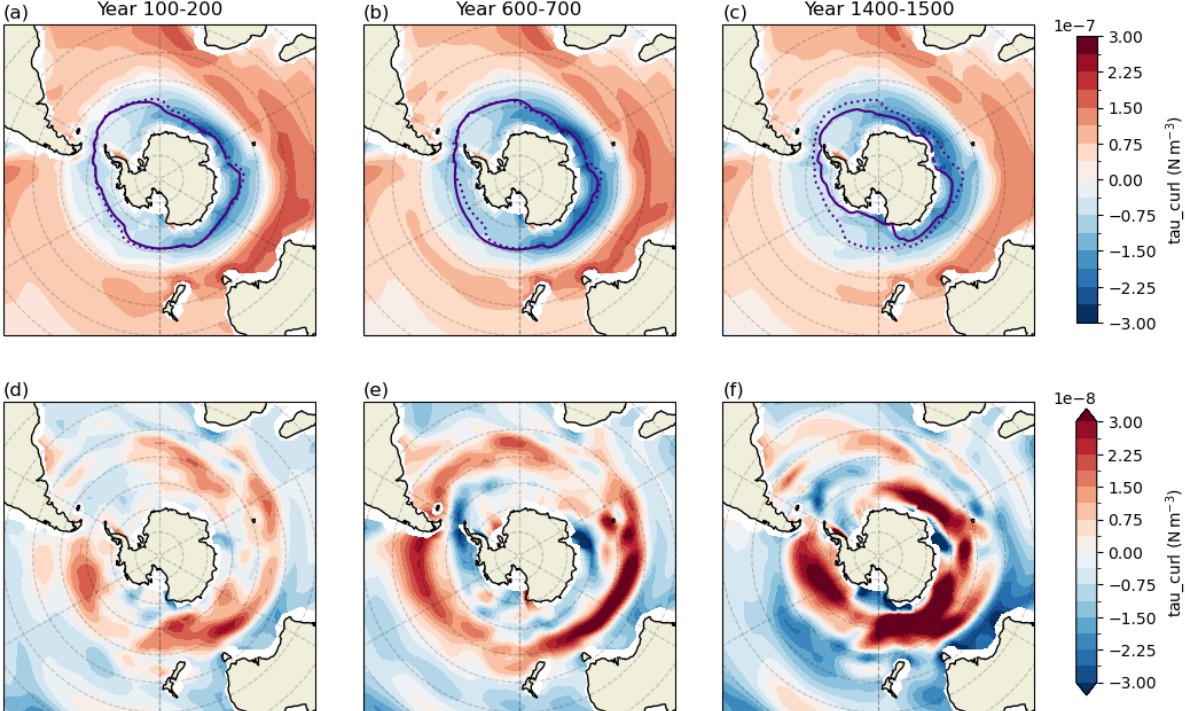

**Figure A6.** Southern ocean (a)-(c) HOSING wind stress curl in $\mathrm{N\,m^{-3}}$ and (d)-(f) anomalies of wind stress curl in $\mathrm{N\,m^{-3}}$. Each column shows a 100 year average for time ranges indicated in Fig. 1. Purple contours in row one show the time averaged maximum sea ice extent where concentration is larger than 15% per grid cell for CONTROL (dotted line) and HOSING (solid line). Latitude graticules are plotted with a 10° grid spacing. Displayed maps (d–f) are similar to subplots f in Figures 3, 5 and 6, but shown here in unified colormap ranges for easy comparison.



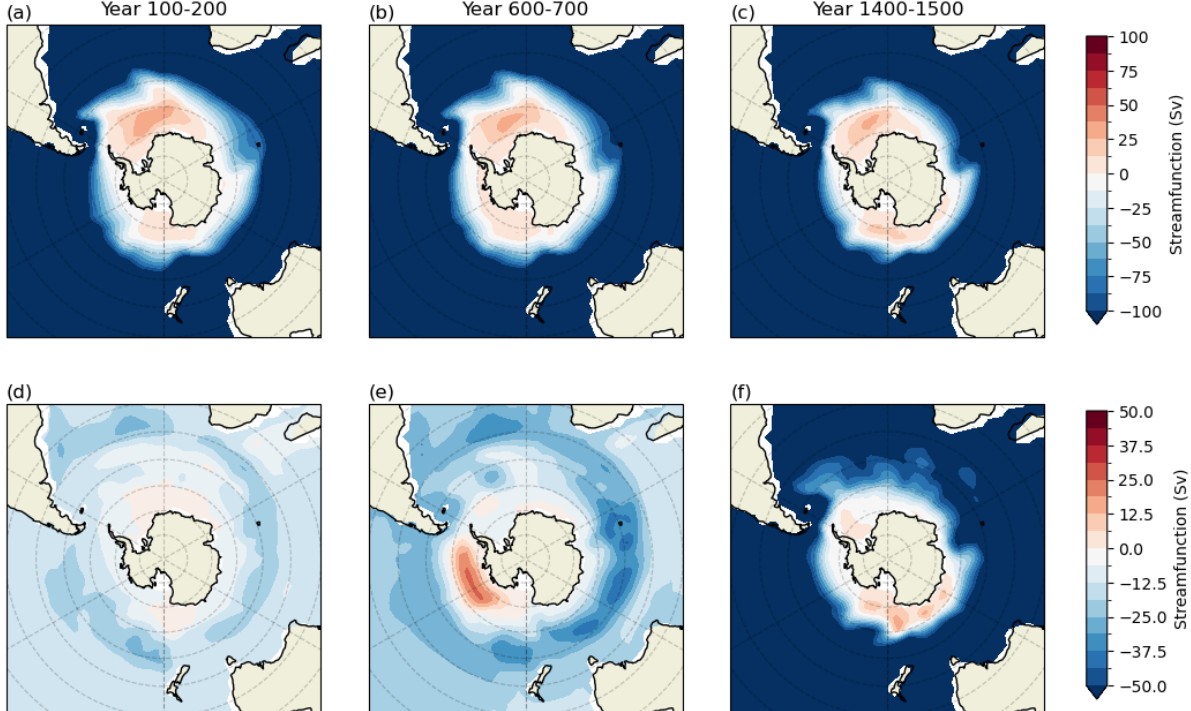

**Figure A7.** Southern ocean (a)-(c) HOSING barotropic stream function in Sv and (d)-(f) anomalies of barotropic stream function in Sv. Each column shows a 100 year average for time ranges indicated in Fig. 1. Purple contours in row one show the time averaged maximum sea ice extent where concentration is larger than 15% per grid cell for CONTROL (dotted line) and HOSING (solid line). Latitude graticules are plotted with a 10° grid spacing.





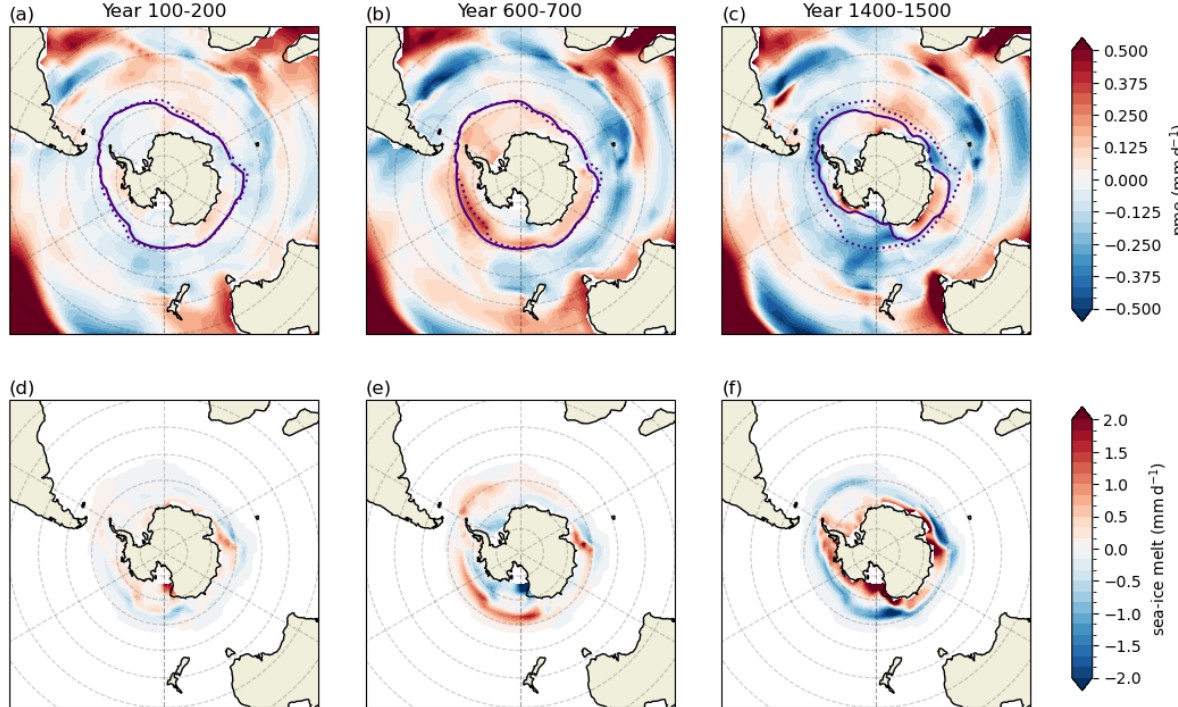

**Figure A8.** Southern ocean anomalies of (a)-(c) precipitation - evaporation in $\mathrm{mm\,d^{-1}}$ and (d)-(f) sea-ice melt freshwater flux in $\mathrm{mm\,d^{-1}}$. Each column shows a 100 year average for time ranges indicated in Fig. 1. Purple contours in row one show the time averaged maximum sea ice extent where concentration is larger than 15% per grid cell for CONTROL (dotted line) and HOSING (solid line). Latitude graticules are plotted with a $10°$ grid spacing.



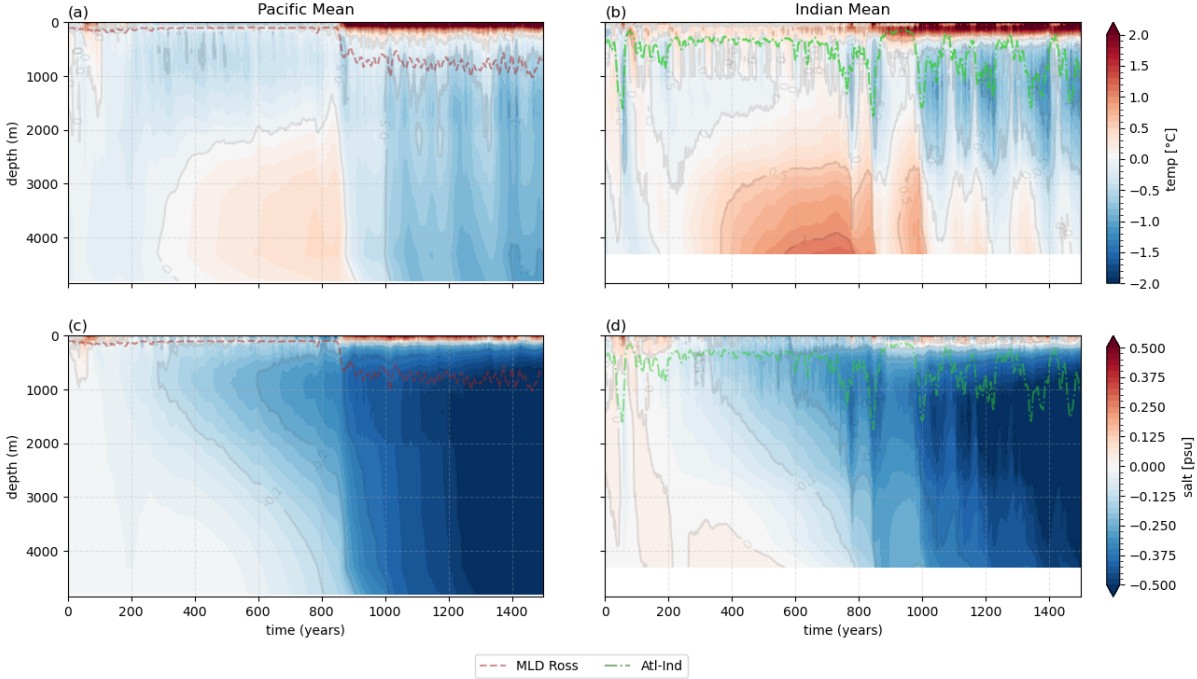

**Figure A9.** Hovmöller plots of (a), (b) temperature and (c), (d) salinity anomalies in °C / psu with respect to a 500-year mean of the CONTROL in the southern hemisphere, south of 65° S. (a) and (c) show averages over the Pacific Sector, (b) and (d) averages over the Indian Ocean Sector. Overlaid time series show the maximum mixed layer depth (MLD) averaged over the Ross Sea (lon=[150° E-150° W], lat<−65° S) (brown, dashed line) and the SO east of the Weddell Sea (Atl-Ind) (lon=[0° E-60° E], lat<−65° S) (green, dash-dotted line).





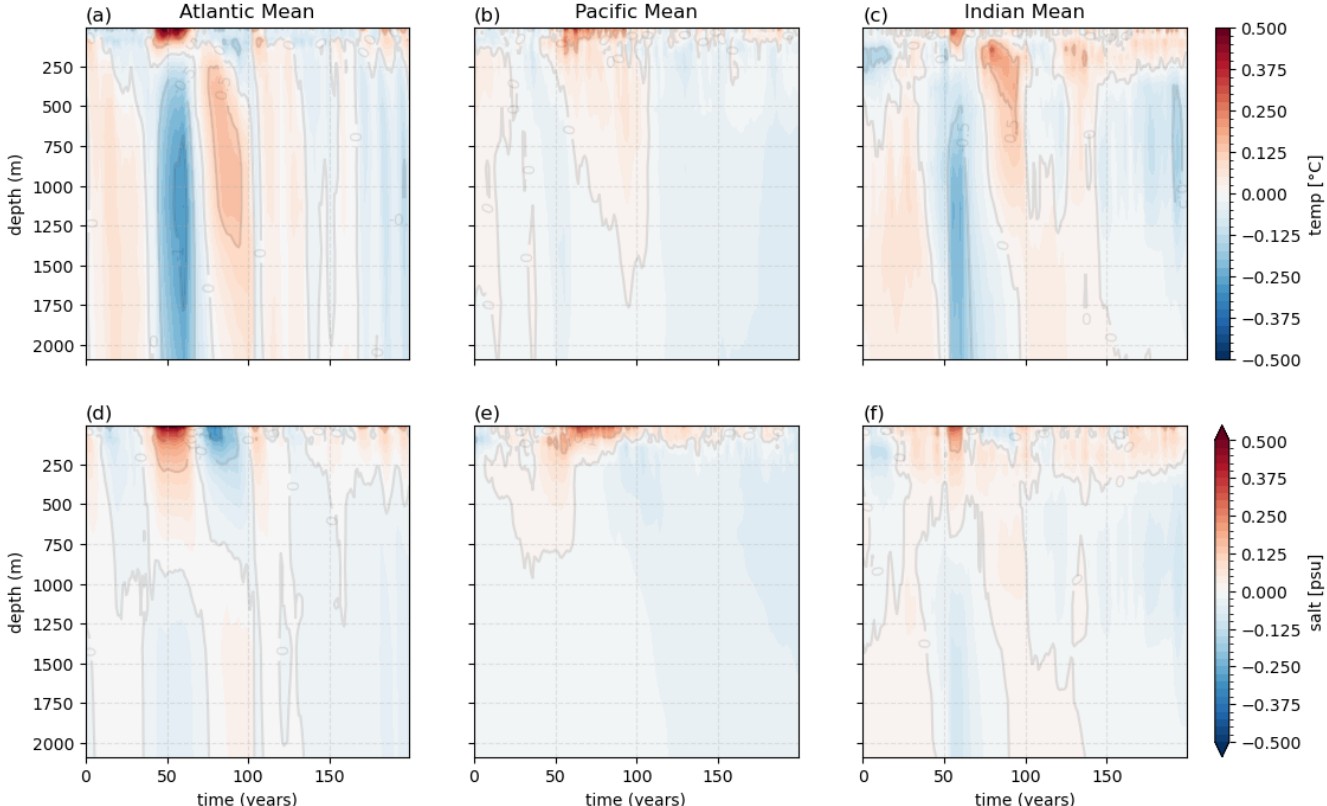

**Figure A10.** Hovmöller plots of (a–c) temperature and (d–f) salinity anomalies in °C / psu with respect to a 500-year mean of the CONTROL in the southern hemisphere, south of 65° S. (a) and (d) show averages over the Atlantic Sector, (b) and (e) show averages over the Pacific Sector, (c) and (f) averages over the Indian Ocean Sector. This Figure show the same fields as Figure 4 / A9, but only the first 200 years of the simulation and limited to the upper 2000 m of the ocean.




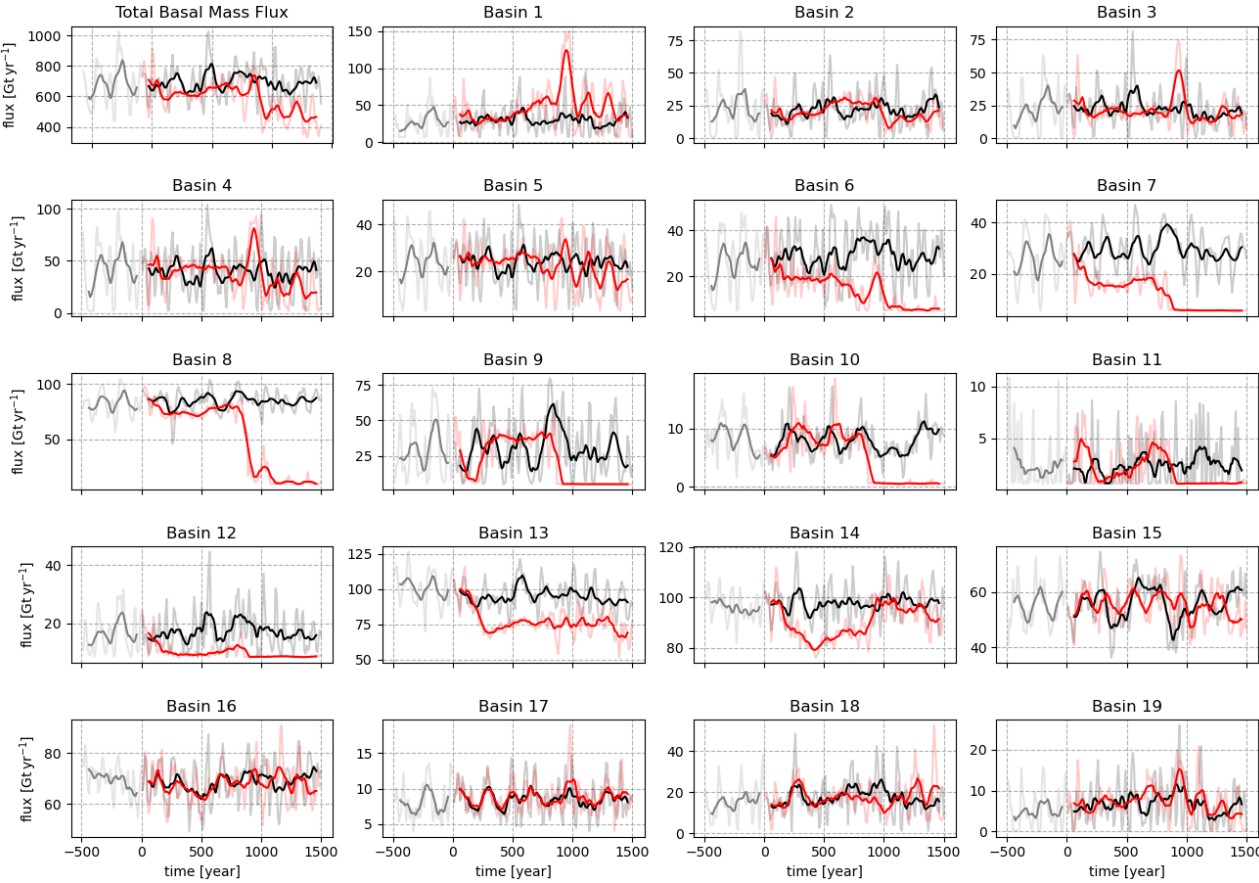

**Figure A11.** Time series of PISM basal mass fluxes in $\mathrm{Gt\,yr^{-1}}$ for CONTROL (black line) and HOSING (red line). Upper left panel shows mean of all basins, each of the other panel the series of the basin (defined as in Reese et al. (2018), Fig. 2) specified in the subtitle. Solid lines show 100 year running mean of the decadal (lighter coloured) data.





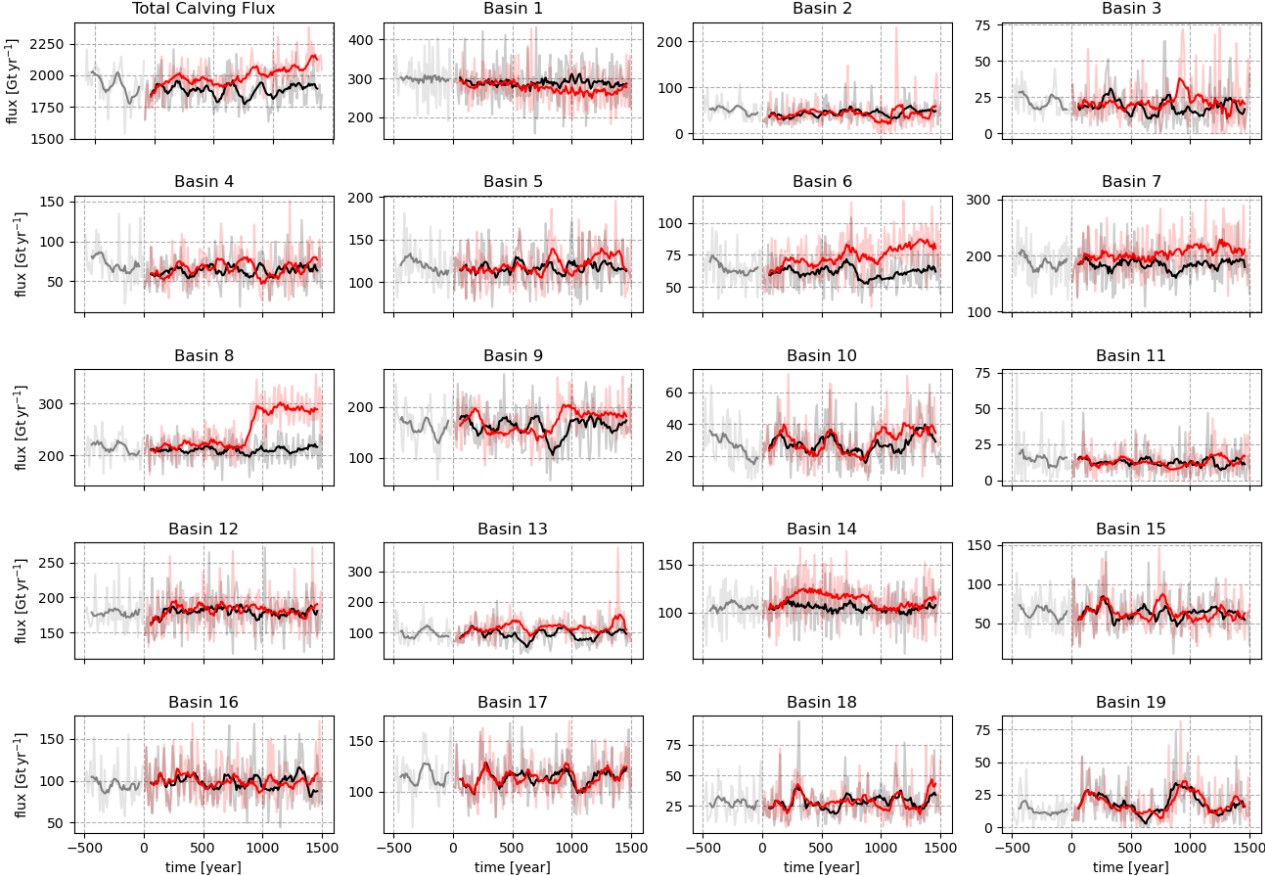

**Figure A12.** Time series of PISM calving fluxes in $\mathrm{Gt\,yr^{-1}}$ for CONTROL (black line) and HOSING (red line). Upper left panel shows mean of all basins, each of the other panel the series of the basin (defined as in Reese et al. (2018), Fig. 2) specified in the subtitle. Solid lines show 100 year running mean of the decadal (lighter coloured) data.





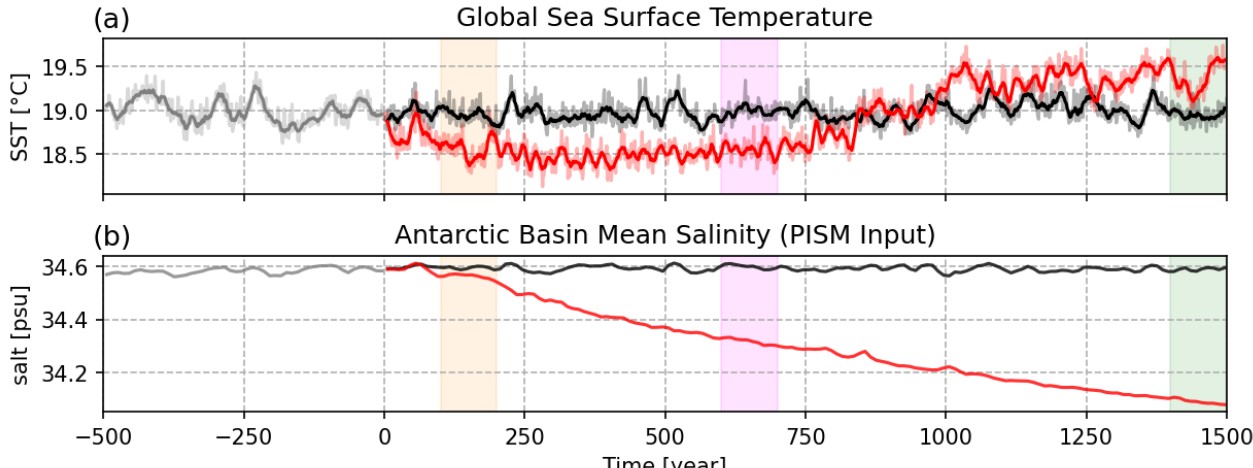

**Figure A13.** Time series of ocean diagnostics: (a) Global mean sea surface temperature in °C and (b) oceanic salinity forcing in psu as the Antarctic basin mean. The CONTROL and HOSING simulations are shown in black and red, respectively, and gray lines show the end of the spinup. The start time of HOSING is set to year 0. In (a) solid lines show 10 year running mean of the yearly (lighter coloured) data. The three shaded areas show the 100 year time periods that are discussed further in section 3.2 – 3.4.



**Appendix B**

This appendix lists the configuration changes against the example configuration CM2M_coarse_BLING as distributed with the MOM5 code, which we applied in our model configuration. The changes were inspired by a discussion in the MOM users forum on google groups (Maroon and Galbraith, 2015). That is not public accessible any more, but relevant parts can be found
in file exp/CM2M_coarse_BLING/README.CM2M_coarse_BLING-PIK in our code publication (Kreuzer, 2025a).

- For the atmosphere model, surface topography information was added. It is missing in the example configuration distributed with MOM5.

- To ensure numerical stability with the added freshwater hosing, it was necessary to halve the timesteps for atmosphere, ocean, and coupler exchange between those two.

- To save comute time, the ocean biogechemistry model BLING was switched off.

- The ocean_basal_tracer module was switched on, in order to enable the insertion of basal melt fluxes at depth.

- `&ocean_sbc_nml/zero_net_water_coupler=.false.` for the hosing run, in order to disable global correction of artificial freshwater flux.

Some parameters were changed from the MOM5 example configuration towards the original settings as used for the CM2Mc publication (Galbraith et al., 2011), to improve model output with respect to preindustrial conditions.
`&ocean_bbc_nml/cdbot` from 1.0e-3 to 2.0e-3
`&ocean_bbc_nml/cdbot_law_of_wall=.false.` was added
`&ocean_nphysics_util_nml/agm_closure_scaling` from 0.12 to 0.1
`&ocean_rivermix_nml/calving_insertion_thickness=40.0` was added
`&ocean_rivermix_nml/runoff_insertion_thickness=40.0` was added
`&ocean_shortwave_gfdl_nml/sw_morel_fixed_depths` from `.true.` to `.false.`
`&ocean_submesoscale_nml/limit_psi_velocity_scale` from 0.10 to 0.50
`&ocean_vert_tidal_nml/shelf_depth_cutoff` from 300.0 to 500.0
`&ocean_vert_tidal_nml/background_diffusivity` from 1.e-5 to 5.e-6

*Author contributions.* Following the CRediT contributor roles Taxonomy: Analysis by AH; Conceptualization by AH, GF, MK, WH; Methodology by AH, GF, MK, SP, WH; Investigation (conducting experiments) by AH, MK; Software by MK, SP; Supervision by GF, MK, WH; Visualization by AH; Writing (original draft) by AH; Writing (Review and Editing) by AH, GF, MK, SP, WH.



*Competing interests.* The authors declare that they have no conflict of interest.

*Acknowledgements.* The authors thank Alexander Robinson for giving feedback and support during the manuscript preparation. Furhtermore, the authors want to thank Stefan Rahmstorf for discussing the methodology and results of the study. The authors also thank Pedro Colombo for collaboration and discussions about the basal melt input at depth implementation in MOM5.

Anna Höse received funding from the European Research Council (ERC Consolidator grant, FORCLIMA, grant no. 101044247). Moritz Kreuzer was financially supported by the Potsdam Graduate School. Willem Huiskamp, as part of PIK's Planetary Boundaries Science Lab, was funded by Virgin Unite. The authors gratefully acknowledge the Ministry of Research, Science and Culture (MWFK) of Land Brandenburg for supporting this project by providing resources on the high performance computer system at the Potsdam Institute for Climate Impact Research. Development of PISM is supported by NASA grants 20-CRYO2020-0052 and 80NSSC22K0274 and NSF grant OAC-2118285.



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
