# Peer review of "Simulating the impact of an AMOC weakening on the Antarctic Ice Sheet using a coupled climate and ice-sheet model"

_EGUsphere, 2025_

## Referee Comment (RC1)

**Review comments for "Simulating the impact of an AMOC weakening on the Antarctic Ice Sheet using a coupled climate and ice-sheet model"**

January 5, 2026

The authors investigate the impact of an AMOC shutdown on the Antarctic Ice Sheet using a global climate model (CM2Mc) interactively coupled to an ice-sheet model (PISM). Overall, the simulated impact on the Antarctic Ice Sheet is limited and predominantly stabilizing, primarily due to decreasing subsurface ocean temperatures around Antarctica that reduce basal melting.

The study is clearly written and represents a welcome contribution to the literature on interacting tipping elements, a field in which destabilizing interactions are more commonly emphasized.

With the exception of the major comment outlined below, which I believe should be addressed before recommending revision, I find the analysis thorough and appropriate, and consider the manuscript well suited to the scope of Earth System Dynamics.

**Major comment**

In this study, as in others that aim to isolate the effects of an AMOC collapse, it is challenging to disentangle the impacts of AMOC weakening from those of the freshwater forcing used to induce it. Achieving an AMOC collapse generally requires the application of a substantial freshwater perturbation in the Northern Hemisphere, which necessitates a choice between conserving global ocean mass and salinity or not, with each approach having inherent drawbacks.

In the present case, the authors choose not to compensate for ocean mass or salinity, applying a freshwater forcing of 0.3 Sv over the North Atlantic and Arctic for 1400 years. This approach has the advantage of avoiding artificial salinification in the Southern Hemisphere. However, the cumulative freshwater input amounts to approximately $1.3 \times 10^{16}$ m$^3$ over the course of the experiment, corresponding to:

- approximately 35 m of global mean sea-level rise;

- about five times the freshwater content of the Greenland Ice Sheet;

- roughly 1% of the total global ocean volume;

- a decrease of about 0.35 psu in global mean ocean salinity if uniformly mixed.

Under these conditions, it becomes difficult to distinguish which fraction of the Southern Ocean freshening arises directly from the imposed freshwater forcing and which results from the AMOC collapse itself, complicating the attribution of the simulated Antarctic Ice Sheet response specifically to AMOC weakening.

Although this issue is acknowledged in the Discussion, I feel that its implications are understated. While much of the analysis focuses on subsurface temperature variations, it is unclear how the substantial changes in ocean mass and salinity affect the interpretation of the results. In particular:

- Is a global mean sea-level rise of ∼35 m truly negligible for Antarctic Ice Sheet dynamics, and especially for the stability of the West Antarctic Ice Sheet?

- The authors acknowledge that the freshwater forcing affects Southern Ocean convection, which appears particularly relevant during the final phase of the experiment (Section 3.4), when most of the freshwater has been released.

- Decreasing subsurface salinity is generally expected to reduce basal melting for a given temperature by increasing the local freezing point and reducing thermal driving. However, in lines 221 and 379, the authors state that the decreasing salinity trend is "not reflected in the AIS basal mass fluxes." It can very well be that correlation with temperatures is more visible because temperatures vary non-linearly, while the salinity changes are approximately linear (Fig A13). No analysis is presented to demonstrate that the contribution of salinity changes to basal melting is negligible.

For these reasons, I believe that the role of the imposed freshwater forcing may be understated in the Discussion. In particular, it remains possible that, in the absence of the large freshwater forcing itself, the subsurface cooling associated with AMOC weakening would suppress basal melting less efficiently, thereby altering the interpretation of the stabilizing Antarctic Ice Sheet response. I therefore recommend that this caveat be explicitly acknowledged, ideally already in the Abstract, and that the Discussion adopt a more balanced treatment of the potential role of the freshwater forcing in shaping the results.

**Specific comments**

I found the results and discussion sections to be well written, and figures to be well presented; my main concern has been addressed in the previous major comment. In what follows, I provide some suggestions for possible modifications in the Abstract and Introduction, as well as a recommendation for an outlook related to the major comment.

**L2** What is meant by "subtropical Southern Ocean"? This terminology is unclear to me.

**L4** I find the expression "grounded ice shelves" misleading. Ice shelves are, by definition, floating.

**L5-7** This sentence reads awkwardly to me, particularly the opening with "As both,". In addition, strictly speaking, the AMOC and the WAIS are not tipping points themselves but tipping elements.

**L16** What is meant by "systematically" in this context?

**L21** Consider replacing this by "the AMOC could substantially weaken or collapse".

**L26** Consider replacing "as well as" with ", and in particular,".

**L46** There appears to be an extra comma here.

**L63–65** This sentence does not read clearly to me and could benefit from rephrasing.

**L90** At a conceptual level with physical models, impacts of an AMOC collapse on the WAIS were investigated by Sinet et al. (2023), especially to capture the slow-fast dynamics of this system. The present study is obviously conducted at a very different level of complexity, and while it does not especially need to be referred to here, I suggest rephrasing the statement to emphasize that such interactions have not previously been explored at this level of complexity.

**L185** While it is reasonable not to fully investigate the case in which a 0.1 Sv freshwater forcing is applied, it would nevertheless be useful to assess whether the qualitative response of the WAIS to AMOC weakening is consistent with the stronger-forcing experiments. In particular because, related to the major comment, the direct impact of the freshwater forcing itself is then reduced.

**Discussion or Conclusion** In relation to the major comment, an additional experiment that could help validate the interpretation would be a similar analysis in which the freshwater flux is compensated to conserve global ocean mass and salinity. I am not suggesting that such an experiment needs to be performed for a revision, but it could be mentioned as a possible outlook or in the Conclusion.